# AMSbench: A Comprehensive Benchmark for Evaluating MLLM Capabilities in AMS Circuits

## Abstract

Analog/Mixed-Signal (AMS) circuits play a critical role in the integrated circuit (IC) industry. However, automating Analog/Mixed-Signal (AMS) circuit design has remained a longstanding challenge due to its difficulty and complexity. Although recent advances in Multi-modal Large Language Models (MLLMs) offer promising potential for supporting AMS circuit analysis and design, current research typically evaluates MLLMs on isolated tasks within the domain, lacking a comprehensive benchmark that systematically assesses model capabilities across diverse AMS-related challenges. To address this gap, we introduce AMSbench, a benchmark suite designed to evaluate MLLM performance across critical tasks including circuit schematic perception, circuit analysis, and circuit design. AMSbench comprises approximately 8000 test questions spanning multiple difficulty levels and assesses eight prominent models, encompassing both open-source and proprietary solutions such as Qwen 2.5-VL and Gemini 2.5 Pro. Our evaluation highlights significant limitations in current MLLMs, particularly in complex multi-modal reasoning and sophisticated circuit design tasks. These results underscore the necessity of advancing MLLMs' understanding and effective application of circuit-specific knowledge, thereby narrowing the existing performance gap relative to human expertise and moving toward fully automated AMS circuit design workflows. Our data is released at this URL.

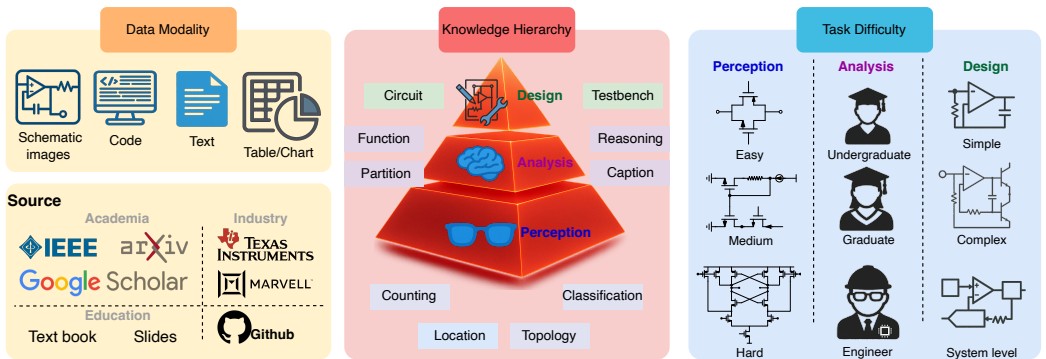

Figure 1: **Overview of AMSbench.** AMSbench includes multimodal question-answer pairs collected from both academia and industry. The tasks are divided into schematic perception, circuit analysis, and circuit design.

## 1 Introduction

The rapid advancement of large language models (LLMs) and multimodal large language models (MLLMs) has led to significant breakthroughs across diverse domains, including autonomous driving (Cui et al., 2024), scientific research (Hao et al., 2025; Yue et al., 2024), mathematics (Zhang et al., 2024; Lu et al., 2023; Yang et al., 2024b), and programming (Zhong & Wang, 2024). In the domain of Electronic Design Automation (EDA), these models have shown promise, particularly in the automated design of digital circuits (Bhandari et al., 2025). On the contrary, automating analog/mixed-signal (AMS) circuit design has been a longstanding challenge for its reliance on

human experience. Today's AI-driven automatic AMS design still faces considerable challenges due to the scarcity of high-quality data and the intrinsic complexity of multi-modal data. As a result, the exploration and application of LLMs in AMS circuit design remain limited and exhibit relatively poor performance (Gao et al., 2025; Lai et al., 2025; Chen et al., 2024). Furthermore, current applications focus on verbal information, while AMS circuits rely on other modalities as well, such as schematics, plots, and charts.

A primary obstacle lies in the limited capability of existing MLLMs to accurately interpret circuit schematics. Unlike netlists, schematics convey richer and more nuanced structural information beyond abstract connectivity. Recent work (Tao et al., 2024; Bhandari et al., 2025) has recognized this limitation and introduced tools capable of automatically converting schematics into netlists, thereby enabling the creation of large-scale, high-quality datasets suitable for training models. Recent advances in the visual capabilities of MLLMs (e.g., GPT-4o (Hurst et al., 2024) and Qwen2.5 (Yang et al., 2024a)) have significantly improved schematic recognition accuracy, laying a solid foundation for the automated analysis and design of AMS circuits. Despite these advancements, current applications often focus on isolated tasks—such as netlist generation (Lai et al., 2025; Liu et al., 2024) and error identification (Chaudhuri et al., 2025)—while lacking comprehensive evaluation frameworks.

In particular, there has been little systematic investigation into the following three fundamental questions:

1. How accurately can models recognize and interpret AMS circuit schematics?

2. What is the upper bound of domain-specific knowledge that models can attain in AMS circuit analysis and design?

3. To what degree are models capable of supporting the automation of AMS circuit design?

To address these questions and bridge the existing research gaps, we propose AMSbench, a comprehensive benchmark designed to evaluate the capabilities of advanced models in the context of AMS circuit design. AMSbench assesses model performance across three key dimensions: **perception**, **analysis**, and **design**.

In the perception task, the objective is to evaluate how accurately MLLMs can generate netlists

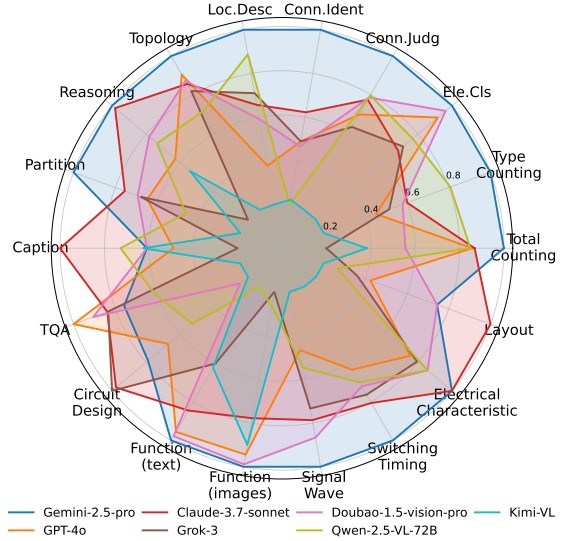

Figure 2: Comparison of top MLLMs on 18 sub-tasks(w/o DeepSeek-R1 on VQA tasks, which lacks visual processing capaility)

directly from circuit schematics, reflecting their schematic recognition capabilities. This is a non-trivial challenge due to the large number of components and their intricate interconnections. We further decompose this task into sub-tasks such as component counting, component classification, and interconnect recognition, culminating in the primary goal of accurate netlist generation. The analysis task examines the models' understanding of circuit-related images, ability to identify critical building blocks, and comprehension of trade-offs among performance metrics—key aspects in AMS circuit design and verification. Finally, the design task investigates whether models can synthesize circuits that satisfy given specifications. We also evaluate their ability to generate appropriate testbenches to assess circuit performance across multiple criteria.

To the best of our knowledge, AMSbench is the first holistic benchmark that systematically evaluates the performance of advanced models in AMS circuits. The overall benchmarking results of state-of-the-art models using AMSbench are illustrated in Fig. 2. Our contributions are summarized as follows:

- We construct **AMSbench**, a large-scale, high-quality multimodal benchmark designed to rigorously evaluate the perception, analysis, and design capabilities of MLLMs in the AMS circuit domain. AMSbench consists of three major components: AMS-Perception (6k), AMS-Analysis (2k), and AMS-Design (68).

- We conduct a comprehensive evaluation of both open-source and proprietary models on AMSbench, providing detailed comparisons and performance insights across all tasks. Furthermore, we present in-depth analyses highlighting the key challenges that must be addressed to enhance the applicability of (M)LLMs in the AMS circuit domain, and we further discuss several potential solutions and research directions to overcome these challenges.

- We release the AMSbench dataset at the provided URL, fostering transparency and reproducibility in this emerging research area.

Table 1: Comparison between existing AMS datasets and benchmarks. *Task* includes three categories: P (Perception), A (Analysis), and D (Design). Q&A stands for Question and Answer.

| Dataset/Benchmark | Modality | Task | Size | Label Type | Difficulty Level |
|---|---|---|---|---|---|
| AMSnet (Tao et al., 2024) | Image-only | P | 1K | Netlist | ✗ |
| Masala-CHI (Bhandari et al., 2025) | Text & Image | P | 6K | Netlist, Caption | ✗ |
| AnalogGenie (Gao et al., 2025) | Text & Image | D | 3K | Netlsit | ✗ |
| Analogcoder (Lai et al., 2025) | Text-only | D | 24 | Netlist | ✔ |
| MMCircuitEval (Zhao et al., 2025) | Text & Image | P&A&D | 3k | Q & A | ✔ |
| **AMSbench(Ours)** | **Text & Image** | **P&A&D** | **8K** | **Netlist, Caption, Q & A** | ✔ |

## 2 RELATED WORK

### 2.1 LLM FOR CIRCUIT DESIGN

LLMs have demonstrated remarkable potential in the field of EDA, excelling in tasks related to system-level design (Yan et al., 2023), RTL (Blocklove et al., 2023; Fu et al., 2023), synthesis and physical design of digital circuits. This success is primarily due to the modular nature of digital circuit descriptions, which resemble software languages. However, AMS circuit designs, with their transistor-level descriptions, pose a significantly greater challenge for LLMs in terms of accurate understanding and description. Some exploratory work has been undertaken in AMS circuit design (Pan et al., 2025; Fang et al., 2025). Artisian (Chen et al., 2024) develops a LLM that automatically generates operational amplifiers by combining advanced prompt engineering techniques like Supervised Fine-Tuning (SFT) and Tree of Thought. Analogcoder (Lai et al., 2025) proposes using LLMs with predefined sub-circuit libraries to achieve an iterative design and optimization flow. AnalogGenie (Gao et al., 2025) converts circuit topologies into Eulerian circuit representations and uses SFT for synthesizing circuits based on the design requirements. To ensure that the generated circuits can meet specifications, AnalogGenie applies Reinforcement Learning with Human Feedback (RLHF) (Ouyang et al., 2022) as a post-training technique. ADO-LLM (Yin et al., 2024) combines LLMs with Bayesian optimization to generate higher-quality candidate design samples, enhancing efficiency in the transistor sizing process. Layout Copilot uses multiple intelligent agents to improve the efficiency and performance of automated layout generation. AMSnet-KG (Shi et al., 2024) employs a knowledge graph-based RAG (Retrieval-Augmented Generation) approach, based on a large-scale, pre-constructed circuit database, to select and generate circuit topologies that meet specifications. However, it is worth noting that these studies mainly focus on purely language-based LLMs, while circuit design often relies heavily on schematic diagrams. Both Masala-CHAI (Bhandari et al., 2025) and AMSnet (Tao et al., 2024) have pointed out that existing MLLMs still lack the capability to effectively recognize circuit schematics.

### 2.2 BENCHMARKING FOR EDA

The academic infrastructure for LLM research in EDA has made significant progress. Abundant available benchmarks and datasets have facilitated effective development of LLMs in EDA. Verilo-gEval (Liu et al., 2023) and RTLLM (Lu et al., 2024) introduce benchmarks for evaluating RTL code generation. However, these benchmarks focus primarily on digital circuits. Due to the complexity and irregularity of analog circuits, AMS circuit design is highly experience-driven, making it difficult to establish fair evaluation methods. Hence, benchmarks in the analog circuit domain remain scarce. We summarize the existing datasets and benchmarks for AMS circuits in Table 1. Analogcoder (Lai et al., 2025) proposes a benchmark to evaluate LLMs in AMS circuit design, categorizing circuits

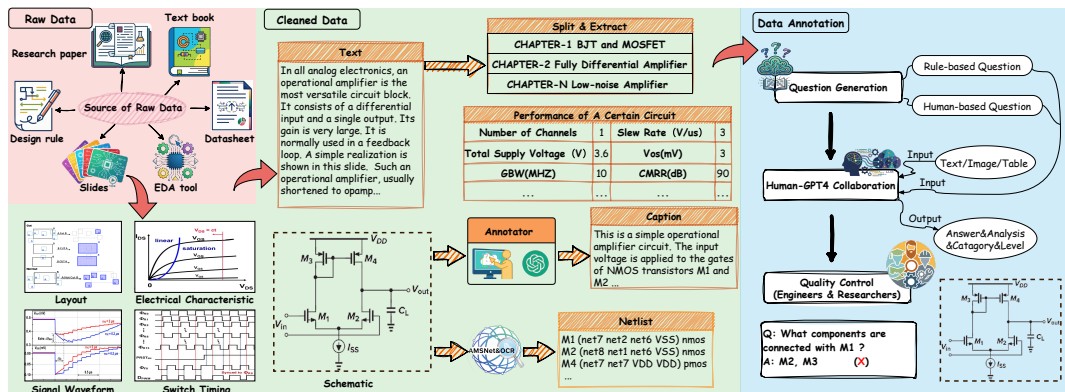

Figure 3: Pipeline of AMSbench construction process.

into three levels: easy, medium, and hard. However, it is limited to the main circuits and did not touch any testbench. Currently, benchmarks in the AMS circuit and EDA domains are limited to verbal questions. However, AMS circuit design is naturally multi-modal, as designers are required to recognize, understand, and reason about circuit schematics.

# 3 AMSBENCH CONSTRUCTION

## 3.1 DATA COLLECTION AND CURATION

To cover a wide range of knowledge and typical question types in the AMS circuit domain, we gather a diverse collection of research papers, textbooks (Razavi, 2005; Gray et al., 2009; Allen & Holberg, 2011; Sansen, 2007), commercial circuit datasheets and EDA tool. We convert all documents from PDF to Markdown format using MinerU (Wang et al., 2024), enabling efficient extraction of embedded visual elements such as circuit schematics. For schematic-to-netlist translation, we utilize AMSnet (Tao et al., 2024) and OCR, which allows us to accurately recover component-level connectivity and circuit topology. To enrich the dataset with semantic information, we use a combination of manual annotations from field experts and outputs from state-of-the-art MLLMs (Hurst et al., 2024; Yang et al., 2024a). We then apply carefully crafted prompt engineering and filter strategies to generate detailed schematic captions. This process yields high-quality pairs of <circuit schematic, caption>. For textbook-derived data, we organize content according to the logical structure and chapter alignment of each source. For datasheet content, we extract structured performance specifications associated with each circuit.

Based on the extracted information, we build a question–answer dataset, where questions are generated through rules and manual design, and answers are obtained from both human experts and LLMs. We adopted a multi-stage data quality control process, relying on professional circuit engineers as well as doctoral and master's students in circuit-related fields to filter and refine the generated data, thereby assisting in the construction and quality assurance of AMSbench, as illustrated in Fig. 4.

## 3.2 EVALUATION

The goal of AMSbench is to thoroughly evaluate MLLMs on the potential applications and tasks in the AMS circuit domain, as shown in Fig. 1. For the design of specific problems, we develop a multi-dimensional evaluation framework that includes **perception**, **analysis**, and **design**. This framework addresses the potential uses of MLLMs in assisting users with interpreting and designing circuit schematics, both automatically and semi-automatically. Considering the complex data modalities and diversity within the AMS circuit domain, the tasks encompass Visual Question Answering (VQA) and Textual Question Answering (TQA). These include multiple-choice questions, computational problems, and open-ended generative questions. We systematically construct questions for each task at multiple levels to accommodate various difficulties and circuit types.

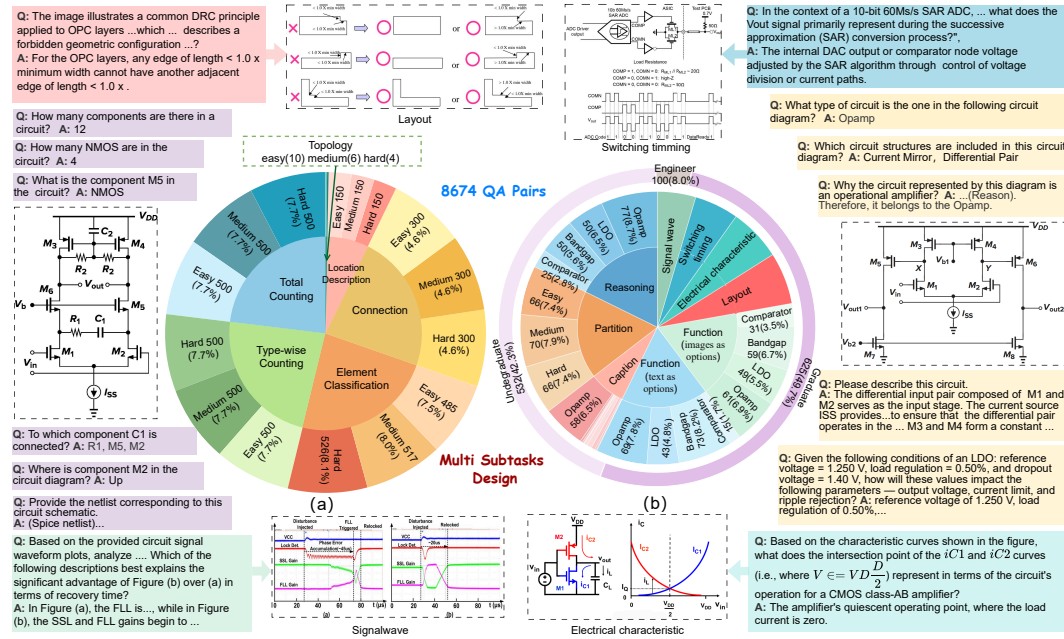

Figure 4: Statistics and examples of AMSBench. Purple regions indicate perception task examples, whereas the others correspond to analysis task examples. (a) Distribution of perception task data. (b) Distribution of analysis task data.

**Evaluation Dimensions** For the **perception** tasks, we focus on recognizing *elements* in circuit schematics. We define an element as any component or device represented by a line in a netlist, such as transistors, resistors, subcircuit symbols, etc. As shown in Fig. 4, MLLMs are evaluated based on three key aspects:

1. Accuracy in Element Counting: This measures how well the model can identify and count the number of different elements in a schematic. We use tasks *type-wise counting* and *element classification*.

2. Precision in Identifying Connectivity: This assesses the model's ability to accurately determine how elements are connected to each other. We use tasks *connection judgment* and *connection identification*, where connection judgment uses true-false questions to decide whether two elements are connected, and connection identification requires the model to state connecting elements.

3. Capability to Recognize the Entire Netlist: This evaluates whether the model can correctly identify the complete netlist of the circuit. We use task *topology*.

Accurate identification of elements, connectivity, and ports is fundamental to understanding and analyzing circuits. The complexity of element types and their connections in schematics makes this task particularly challenging. It requires the MLLM to have a more rigorous perception capability compared to traditional visual counting tasks.

The **analysis** tasks in AMSbench primarily assess the MLLMs' comprehension of circuit schematics. This includes recognizing and analyzing the functions of circuits, as well as identifying key functional building blocks within them, as illustrated in Fig. 4. Beyond schematic understanding, the analysis tasks also cover other critical aspects of circuit design, such as interpreting signal waveforms, evaluating switching timing, and analyzing layout-related information. Additionally, we evaluate the understanding of AMS circuits, such as trade-offs between different circuit performance metrics by both LLMs and MLLMs. Accurately analyzing a circuit and its

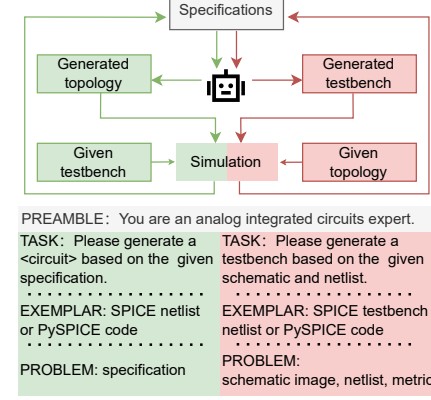

Figure 5: Design task flow

corresponding performance metrics is the foundation for ensuring accurate and effective circuit design.

The **design** tasks in our study consider both circuit design and testbench design, as shown in Fig. 5. Proper circuit design ensures that the functionality meets specifications, while effective testbench design guarantees that the circuit's performance can be accurately measured and validated. These two tasks are central to the AI-driven automation of AMS circuit design. In setting up the circuit design tasks, we adopt and expand upon the benchmark established by AnalogCoder (Lai et al., 2025).

**Difficulty Levels**    We classify the questions into three difficulty levels. Specifically, for the **perception** task, we categorize the difficulty based on the number of elements in the circuit: simple (num < 9), medium ( 9< num < 16), and difficult (num > 16). For circuit functionality **analysis**, we classify the problems according to the circuit type and group them into two levels based on their appearance in educational stages: undergraduate and graduate levels. For testing the trade-offs between circuit performances, we assign a classification suitable for engineers. For the **design** task, we classify the circuits based on their complexity into three levels: simple, complex, and system-level circuits.

Table 2: Circuit design tasks. Number of (simple / complex / system-level) tasks are shown for each circuit type.

| Circuit Type | # of Tasks | Circuit Type | # of Tasks | Circuit Type | # of Tasks | Circuit Type | # of Tasks |
|---|---|---|---|---|---|---|---|
| Amplifier | 7 / 0 / 0 | Oscillator | 0 / 2 / 0 | Subtractor | 0 / 1 / 0 | LDO | 0 / 1 / 0 |
| Inverter | 2 / 0 / 0 | Integrator | 0 / 1 / 0 | Schmitt trigger | 0 / 1 / 0 | Comparator | 0 / 1 / 0 |
| Current mirror | 2 / 0 / 0 | Differentiator | 0 / 1 / 0 | VCO | 0 / 1 / 0 | PLL | 0 / 0 / 1 |
| Op-amp | 2 / 2 / 0 | Adder | 0 / 1 / 0 | Bandgap | 0 / 1 / 0 | SAR-ADC | 0 / 0 / 1 |

Table 3: Testbench design tasks with number of metrics required per testbench suite.

| ID | Circuit Type | # of Metrics | ID | Circuit Type | # of Metrics | ID | Circuit Type | # of Metrics |
|---|---|---|---|---|---|---|---|---|
| 1 | Cross-coupled differential amplifier | 7 | 5 | PLL | 2 | 9 | Unit capacitor | 1 |
| 2 | Comparator | 2 | 6 | MOS_Ron | 1 | 10 | Folded cascode OTA | 5 |
| 3 | Bootstrap | 1 | 7 | LDO | 7 | 11 | SAR-ADC | 1 |
| 4 | Telescopic cascode OTA | 7 | 8 | VCO | 2 | 12 | Bandgap | 4 |

## 3.3    AMSBENCH STATISTICS

Fig. 4(a) illustrates the subtasks involved in the perception task along with the number of questions at varying difficulty levels. Fig. 4(b) presents statistical information for the analysis task and its various subtasks. The VQA tasks focus on evaluating the MLLM's ability to interpret circuit-related images, while the TQA tasks assess the model's understanding of circuit knowledge and its awareness of performance trade-offs. Table 2 and 25 present an overview of the design tasks for circuits and testbenches, respectively. For the circuit design section, we incorporated the benchmarks provided by AnalogCoder (Lai et al., 2025) and further extended them with additional circuit tasks, including system-level circuit design. The testbench design tasks address a notable gap in the current community by introducing a previously underexplored category.

## 4    EXPERIMENTS

### 4.1    MODELS

We perform experiments on mainstream closed-source MLLMs: GPT-4o (Hurst et al., 2024), Grok-3 (gro, 2025), Gemini-2.5-pro (Team et al., 2023), Claude3.7 sonnet (Anthropic, 2024), Doubao-1.5-vision-pro-32k (Guo et al., 2025b), and powerful open-source models: Kimi-VL (Team et al., 2025), Qwen2.5-VL 72B (Yang et al., 2024a), DeepSeek-R1 (Guo et al., 2025a). We evaluate both TQA tasks on all models, and VQA tasks on all models except DeepSeek-R1. We use all open-source models with default parameters and deploy on up to 8 A100 GPUs.

### 4.2    METRICS

For multiple-choice questions, we adopt accuracy (ACC) as the evaluation metric. For multiple-selection questions, we use the F1 score. For netlist recognition tasks, we define a Netlist Edit

Distance (NED) as the evaluation metric, with the calculation procedure illustrated in Fig. 6. The NED for each schematic image is normalized as shown in (1):

$$\text{NED}_{norm} = \frac{|\,\text{GT} \cup \text{Pred}\,| - |\,\text{GT} \cap \text{Pred}\,|}{|\,\text{GT}\,|} \qquad (1)$$

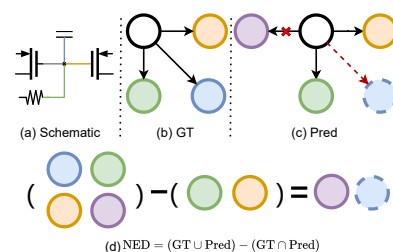

(a) Schematic    (b) GT    (c) Pred

(d) NED = (GT ∪ Pred) − (GT ∩ Pred)

Figure 6: Edit distance computation between the GT and the predicted netlist.

For evaluating the circuit design tasks, we use pass@$k$ as the primary metric to measure the success rate of model-generated circuits. Syntax@$k$ and Metric@$k$ are employed to evaluate the generated testbenches. Syntax@$k$ examines the presence of syntactic errors that hinder simulation, and Metric@$k$ verifies the functional correctness of the corresponding test circuits.

### 4.3 EXPERIMENTAL RESULTS

**Perception Tasks:** Table 4 presents the models' performance on fundamental circuit schematic recognition tasks. Specifically, component counting and classification, both of which are essential for accurate netlist extraction. Gemini achieves the best overall results. However, due to the complexity and diversity of component types, the models show limitations in accurate counting. For element type classification, Gemini performs well, reaching 94% accuracy. Among open-source models, Qwen2.5-VL achieves 86%, suggesting that open-source models still have room for improvement in component type recognition.

The lower part of Table 4 presents the accuracy of MLLMs in identifying inter-device connectivity. While the models can produce reasonably accurate predictions for local connections, they fall short in reconstructing the complete netlist. Even netlists produced by the best-performing model, Gemini, require substantial modifications to align with the ground truth. Closed-source models perform significantly better on these tasks, whereas Kimi-VL fail to produce outputs in the correct format.

Table 4: Comprehensive comparison of models across perception tasks and circuit interconnect recognition. Abbreviations adopted: Ele. Cls = Element Classification, Loc. Desc = Location Description, Conn. Judg = Connection Judgment, Conn. Ident = Connection Identification.

| Models | Total Counting | Total Counting | Type Counting | Type Counting | Ele. Cls |
| --- | --- | --- | --- | --- | --- |
| | ACC (↑) | MSE (↓) | ACC (↑) | MSE (↓) | ACC (↑) |
| Gemini-2.5-pro | **0.65** | **10.02** | **0.64** | **13.41** | **0.94** |
| GPT-4o | 0.51 | 19.05 | 0.54 | 28.18 | 0.91 |
| Claude-3.7-sonnet | 0.36 | 18.38 | 0.55 | 24.18 | 0.83 |
| Grok-3 | 0.22 | 60.71 | 0.50 | 26.48 | 0.84 |
| Doubao-1.5-vision-pro | 0.24 | 38.13 | 0.51 | 24.76 | 0.93 |
| Kimi-VL-A3B | 0.15 | 49.19 | 0.44 | 34.96 | 0.66 |
| Qwen2.5-VL-72B | 0.43 | 19.59 | 0.49 | 18.59 | 0.86 |

| Models | Loc. Desc | Conn. Judg | Conn. Ident | Topology | – |
| --- | --- | --- | --- | --- | --- |
| | ACC (↑) | ACC (↑) | F1 (↑) | NED (↓) | – |
| Gemini-2.5-pro | **0.61** | **0.85** | **0.88** | **0.91** | – |
| GPT-4o | 0.37 | 0.73 | 0.65 | 1.40 | – |
| Claude-3.7-sonnet | 0.48 | 0.76 | 0.71 | 1.65 | – |
| Grok-3 | 0.50 | 0.70 | 0.65 | 1.84 | – |
| Doubao-1.5-vision-pro | 0.45 | 0.76 | 0.64 | 1.57 | – |
| Kimi-VL-A3B | 0.31 | 0.53 | 0.53 | – | – |
| Qwen2.5-VL-72B | 0.56 | 0.77 | 0.52 | 2.38 | – |

**Analysis Tasks:** Table 5 summarizes the models' capabilities of analyzing AMS circuits. In schematic interpretation(*Reasoning*, *Partition*, *Caption*, *Function*), different MLLMs exhibit distinct strengths: Gemini demonstrates the highest accuracy in identifying and analyzing functional building blocks, while Claude-Sonnet provides more accurate overall descriptions of circuit behavior. Gemini also demonstrated strong performance on other circuit-related images. Table 16 shows that current models can achieve relatively high accuracy in analyzing circuit knowledge designed for undergraduate and graduate education. However, they perform poorly in understanding the trade-offs between

Table 5: Comparison of models on analysis tasks

| Models | Reasoning | Signal wave | Switching timing | Electrical characteristic | Layout |
|---|---|---|---|---|---|
| | ACC (↑) | ACC (↑) | ACC (↑) | ACC (↑) | ACC (↑) |
| Gemini-2.5-pro | **0.92** | **0.94** | **0.96** | **0.92** | 0.75 |
| GPT-4o | 0.77 | 0.74 | 0.79 | 0.80 | 0.65 |
| Claude-3.7-sonnet | 0.91 | 0.86 | 0.87 | **0.92** | **0.83** |
| Grok-3 | 0.61 | 0.84 | 0.85 | 0.82 | 0.63 |
| Doubao-1.5-vision-pro | 0.83 | 0.89 | 0.83 | 0.85 | 0.75 |
| Kimi-VL-A3B | 0.74 | 0.64 | 0.59 | 0.53 | 0.58 |
| Qwen2.5-VL-72B | 0.82 | 0.77 | 0.82 | 0.85 | 0.60 |
| **Models** | **Partition** | **Caption** | **Function (text)** | **Function (image)** | **TQA** |
| | F1 (↑) | ACC (↑) | ACC (↑) | ACC (↑) | ACC (↑) |
| Gemini-2.5-pro | **0.80** | 0.70 | **0.95** | **0.94** | 0.72 |
| GPT-4o | 0.57 | 0.61 | 0.93 | 0.89 | **0.78** |
| Claude-3.7-sonnet | 0.64 | **0.98** | 0.88 | 0.74 | 0.74 |
| Grok-3 | 0.59 | 0.41 | 0.77 | 0.22 | 0.74 |
| Doubao-1.5-vision-pro | 0.60 | 0.70 | 0.94 | 0.93 | 0.76 |
| Kimi-VL-A3B | 0.25 | 0.71 | 0.59 | 0.28 | 0.59 |
| Qwen2.5-VL-72B | 0.45 | 0.78 | 0.78 | 0.85 | 0.69 |

circuit performance metrics commonly encountered in industry. Even the best-performing model, GPT-4o, only achieves 58% accuracy, indicating that LLMs currently lack a clear understanding of the expected performance characteristics of each circuit in the design process.

**Design Tasks:** Table 6 shows the performance of the models in circuit design and testbench design tasks. For circuit design, Grok-3 and Claude-Sonnet achieve the best results. However, for testbench design, none of the current models can directly generate syntactically correct testbenches, with only occasionally exceptions of GPT-4o. One possible reason is that the current pretraining data lacks sufficient testbench-related knowledge. Additionally, the metrics that need to be measured vary across different circuits, making testbench generation highly challenging.

Table 6: Comparison of models and circuit design and testbench design tasks. The data presents the average results of all the circuits listed in Table 2. Detailed results are available in the appendix in Tables 18-27. *Syntax*: generated testbench is syntactically correct to run simulation. *Metric*: generated testbench is topologically and parametrically correct and produces the correct performance metric.

| Model | Circuit Design | | | Testbench Design | |
|---|---|---|---|---|---|
| | Pass@3 | Pass@5 | Pass@10 | Syntax@5 | Metric@5 |
| Gemini-2.5-pro | 0.57 | 0.54 | 0.43 | 0 | 0 |
| GPT-4o | 0.47 | 0.49 | 0.42 | **0.084** | 0 |
| Claude-3.7-sonnet | 0.63 | **0.64** | 0.50 | 0 | 0 |
| Grok-3 | **0.65** | 0.54 | **0.61** | 0 | 0 |
| Doubao-1.5-vision-pro | 0.45 | 0.24 | 0.15 | 0 | 0 |
| Qwen2.5-VL-72B | 0.47 | 0.41 | 0.33 | 0 | 0 |
| Kimi-VL-A3B | 0.41 | 0.25 | 0.13 | 0 | 0 |
| DeepSeek-R1 | 0.55 | 0.51 | 0.45 | - | - |

## 5 CHALLENGES AND POSSIBLE IMPROVEMENTS

**Challenges of MLLMs in the interpretation of circuit-related images.** Existing MLLMs remain limited in accurately interpreting circuit schematics. Although some models can capture localized connectivity patterns, their performance degrades when extracting complete netlists. A key challenge lies in the domain gap between circuit schematics and the natural images typically used in MLLM pretraining, which leads to misclassification of components—for example, failing to distinguish between PMOS and NMOS transistors. In addition, MLLMs often struggle to correctly associate pins and ports with their parent components, resulting in connectivity errors.

To enhance model's recognition capability, one approach is to combine MLLMs with specialized vision models such as object detection (Bhandari et al., 2025) or OCR, which improves baseline

component recognition but does not fundamentally advance the MLLMs themselves. A more promising direction is the construction of large-scale circuit-specific multimodal datasets, enabling continual pretraining and post-training. We have conducted experiments on component grounding, and the results show that MLLMs trained with such data exhibit improved perception of circuit schematics, as illustrated in the Fig. 37.

**Hallucinations in circuit analysis.** In circuit analysis tasks, one major reason for poor performance lies in the inability of the vision encoder to accurately and comprehensively embed visual information, as discussed earlier. Another reason is the insufficient circuit-related knowledge of the models themselves, which is also evident in text-only evaluation tasks. In existing works that combine LLMs with AMS circuit design, LLMs are typically employed as analysis tools for downstream design (Yin et al., 2024; Wei et al., 2025). A model with strong circuit analysis capabilities can significantly reduce the parameter search space and partition the circuit into macros, thereby enabling efficient layout generation. However, current models still lack the ability to accurately quantify the trade-offs between circuit performance metrics, which limits their capability to recommend appropriate circuit topologies under given target specifications.

One possible solution to enhance model performance in analysis tasks is to construct high-quality datasets for training. However, due to the scarcity of documents related to circuit analysis and design, training outcomes cannot be guaranteed. Another promising approach is to adopt Retrieval-Augmented Generation (RAG), which dynamically collects high-quality multimodal data, cleans it, and stores it in a vector database. During circuit analysis, the model can then retrieve relevant knowledge from this database to support its reasoning and responses.

**Struggles in AMS circuit design.** Applying LLMs to AMS design involves three key stages: topology design, testbench generation, and circuit sizing. End-to-end generation of directly usable circuits remains highly challenging. For **topology design**, the primary requirement is to generate novel yet functional circuits. Possible solutions include continual training of LLMs, combined with prompt engineering. However, for complex and system-level circuits, directly producing a correct design remains extremely challenging, as illustrated in Fig. 35. A divide-and-conquer strategy, where submodules are designed and validated individually before integration, offers a more feasible path. For **testbench generation**, the process is relatively standardized. Although it is still difficult to directly generate usable testbenches, one feasible approach is leveraging predefined libraries and letting LLMs perform targeted modifications. For **circuit sizing**, it is extremely difficult for an LLM to directly produce circuits with desired sizing. One potential solution is to combine LLMs with traditional black-box optimization algorithms, embedding domain knowledge into the optimization process to accelerate convergence. Another promising approach is the use of multi-agent systems, where LLMs emulate the workflow of human engineers. The former approach enables exploration of a broader design space, albeit at the expense of efficiency, while the latter offers improved interpretability and transparency, but the accumulation of hallucinations from each agent may negatively affect the final performance.

## 6 CONCLUSION

This paper introduces AMSbench, a benchmark designed to evaluate the capabilities of MLLMs in the AMS circuit domain. The benchmark comprehensively assesses model performance across three key dimensions—schematic perception, circuit analysis, and circuit design—covering a variety of tasks. AMSbench reveals significant limitations in current MLLMs, especially in schematic perception and complex circuit design. While certain models perform adequately in basic component recognition and simpler circuit analysis tasks, they notably struggle with advanced tasks, including accurate schematic interpretation and system-level circuit design. Given the increasing interest in applying MLLMs to automate AMS design processes, AMSbench provides an essential evaluation framework, establishing a robust foundation for future advancements in this field. Achieving high-performance scores on AMSbench would signify substantial progress and tangible benefits in the automation of AMS circuit design. Future research will prioritize the expansion of datasets to enhance the robustness and generalizability of multimodal models. It will investigate advanced methodologies, such as RAG and RLHF, to augment design capabilities.

## REPRODUCIBILITY STATEMENT

We provide the complete examples of AMSbench in the Appendix C, and release the full evaluation dataset at the provided anonymous URL to facilitate reproducibility.

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
