# A  THE USE OF LARGE LANGUAGE MODELS (LLMS)

This research employed multiple large language models as test subjects for benchmark development and validation. Additionally, LLMs were used minimally for sentence-level language refinement and polishing to improve clarity of expression. All core research contributions, methodology, and analysis were conducted independently by the authors without AI assistance.

# B  DETAILED EXPERIMENT RESULTS

## B.1  RESULTS OF PERCEPTION TASKS

| Models | Metric | Counting Task | | | | | | | |
| --- | --- | --- | --- | --- | --- | --- | --- | --- | --- |
| | | Total Counting | | | | Type-wise Counting | | | |
| | | Easy | Medium | Hard | Total | Easy | Medium | Hard | Total |
| Gemini-2.5-pro | ACC (↑) | **0.88** | **0.65** | **0.45** | **0.65** | **0.83** | 0.63 | 0.46 | 0.64 |
| | MSE (↓) | **0.39** | **2.30** | **27.36** | **10.02** | **0.67** | **3.75** | 35.81 | 13.41 |
| GPT-4o | ACC (↑) | 0.75 | 0.51 | 0.26 | 0.51 | 0.76 | 0.57 | 0.30 | 0.54 |
| | MSE (↓) | 0.80 | 2.63 | 53.72 | 19.05 | 2.15 | 6.96 | 75.44 | 28.18 |
| Claude-3.7-sonnet | ACC (↑) | 0.72 | 0.27 | 0.01 | 0.36 | 0.73 | 0.57 | **0.35** | **0.55** |
| | MSE (↓) | 0.56 | 3.63 | 50.96 | 18.38 | 2.29 | **6.67** | 63.57 | 24.18 |
| Grok-3 | ACC (↑) | 0.47 | 0.14 | 0.04 | 0.22 | 0.70 | 0.53 | 0.27 | 0.50 |
| | MSE (↓) | 3.58 | 21.50 | 157.05 | 60.71 | 2.49 | 6.92 | 68.51 | 26.48 |
| Doubao-1.5-vision-pro | ACC (↑) | 0.60 | 0.10 | 0.01 | 0.24 | 0.71 | 0.53 | 0.30 | 0.51 |
| | MSE (↓) | 0.67 | 5.93 | 107.78 | 38.13 | 2.56 | 9.14 | **62.59** | **24.76** |
| Kimi-VL-A3B | ACC (↑) | 0.23 | 0.13 | 0.08 | 0.15 | 0.63 | 0.46 | 0.23 | 0.44 |
| | MSE (↓) | 15.68 | 28.04 | 103.86 | 49.19 | 2.86 | 13.44 | 94.38 | 34.96 |
| Qwen2.5-VL-72B | ACC (↑) | 0.67 | 0.42 | 0.21 | 0.43 | 0.68 | 0.47 | 0.31 | 0.49 |
| | MSE (↓) | 1.09 | 3.98 | **53.90** | **19.59** | 1.87 | **6.55** | **47.35** | **18.59** |

Table 7: Counting Task Performance Across Different Models

| Models | Element Classification Task (ACC ↑) | | | |
| --- | --- | --- | --- | --- |
| | Easy | Medium | Hard | Total |
| Gemini-2.5-pro | **0.96** | **0.94** | **0.91** | **0.94** |
| GPT-4o | 0.93 | 0.93 | 0.88 | 0.91 |
| Claude-3.7-sonnet | 0.90 | 0.87 | 0.84 | 0.83 |
| Grok-3 | 0.84 | 0.84 | 0.83 | 0.84 |
| Doubao-1.5-vision-pro | 0.94 | 0.93 | **0.91** | 0.93 |
| Kimi-VL-A3B | 0.69 | 0.65 | 0.63 | 0.66 |
| Qwen2.5-VL-72B | 0.88 | 0.86 | 0.83 | 0.86 |

Table 8: Element Classification Task Performance Across Different Models (in decimals)

| Models | Metric | Interconnect Perception Task | | | | | | | |
| --- | --- | --- | --- | --- | --- | --- | --- | --- | --- |
| | | Connection Judgment | | | | Connection Identification | | | |
| | | Easy | Medium | Hard | Total | Easy | Medium | Hard | Total |
| Gemini-2.5-pro | ACC (↑) | **0.86** | **0.83** | **0.85** | **0.85** | **0.78** | **0.51** | **0.50** | **0.60** |
| | F1 (↑) | - | - | - | - | **0.93** | **0.86** | **0.86** | **0.88** |
| GPT-4o | ACC (↑) | 0.74 | 0.71 | 0.74 | 0.73 | 0.27 | 0.12 | 0.10 | 0.16 |
| | F1 (↑) | - | - | - | - | 0.74 | 0.61 | 0.59 | 0.65 |
| Claude-3.7-sonnet | ACC (↑) | 0.74 | 0.76 | 0.77 | 0.76 | 0.36 | 0.25 | 0.22 | 0.27 |
| | F1 (↑) | - | - | - | - | 0.74 | 0.71 | 0.68 | 0.71 |
| Grok-3 | ACC (↑) | 0.72 | 0.72 | 0.67 | 0.70 | 0.25 | 0.16 | 0.13 | 0.18 |
| | F1 (↑) | - | - | - | - | 0.70 | 0.63 | 0.62 | 0.65 |
| Doubao-1.5-vision-pro | ACC (↑) | 0.78 | 0.74 | 0.77 | 0.76 | 0.30 | 0.13 | 0.14 | 0.19 |
| | F1 (↑) | - | - | - | - | 0.72 | 0.60 | 0.59 | 0.64 |
| Kimi-VL-A3B | ACC (↑) | 0.55 | 0.54 | 0.51 | 0.53 | 0.13 | 0.09 | 0.10 | 0.10 |
| | F1 (↑) | - | - | - | - | 0.57 | 0.51 | 0.52 | 0.53 |
| Qwen2.5-VL-72B | ACC (↑) | 0.75 | 0.78 | 0.77 | 0.77 | 0.20 | 0.10 | 0.13 | 0.14 |
| | F1 (↑) | - | - | - | - | 0.56 | 0.48 | 0.53 | 0.52 |

Table 9: Interconnect Perception Task Performance Across Different Models

| Models | Location Description Task (ACC ↑) | | | |
| --- | --- | --- | --- | --- |
| | Easy | Medium | Hard | Total |
| Gemini-2.5-pro | **0.61** | **0.60** | **0.61** | **0.61** |
| GPT-4o | 0.44 | 0.31 | 0.37 | 0.37 |
| Claude-3.7-sonnet | 0.49 | 0.45 | 0.50 | 0.48 |
| Grok-3 | 0.45 | 0.55 | 0.49 | 0.50 |
| Doubao-1.5-vision-pro | 0.51 | 0.47 | 0.38 | 0.45 |
| Kimi-VL-A3B | 0.32 | 0.33 | 0.27 | 0.31 |
| Qwen2.5-VL-72B | 0.57 | 0.59 | 0.53 | 0.56 |

Table 10: Location Description Task Performance Across Different Models

| Models | Topology Generation Task (NED ↓) | | | |
| --- | --- | --- | --- | --- |
| | Easy | Medium | Hard | Total |
| Gemini-2.5-pro | **0.73** | **0.95** | **1.34** | **0.91** |
| GPT-4o | 1.31 | 1.56 | 1.46 | 1.40 |
| Claude-3.7-sonnet | 1.72 | 1.64 | 1.51 | 1.65 |
| Grok-3 | 2.08 | 1.71 | 1.37 | 1.84 |
| Doubao-1.5-vision-pro | 1.54 | 1.52 | 1.68 | 1.57 |
| Kimi-VL-A3B | – | – | – | – |
| Qwen2.5-VL-72B | 2.65 | 2.14 | 1.95 | 2.38 |

Table 11: Topology Generation Task Performance Across Different Models

## B.2 RESULTS OF ANALYSIS TASKS

| Models | Function Identification Task (ACC ↑) | |
|---|---|---|
| | Function (text as options) | Function (image as options) |
| Gemini-2.5-pro | **0.95** | **0.94** |
| GPT-4o | 0.93 | 0.89 |
| Claude-3.7-sonnet | 0.88 | 0.74 |
| Grok-3 | 0.77 | 0.22 |
| Doubao-1.5-vision-pro | 0.94 | 0.93 |
| Kimi-VL-A3B | 0.59 | 0.28 |
| Qwen2.5-VL-72B | 0.78 | 0.85 |

Table 12: Function Identification Task Performance Across Different Models

| Models | Circuit Partition Task | | | | | | | |
|---|---|---|---|---|---|---|---|---|
| | Easy | | Medium | | Hard | | Overall | |
| | ACC ↑ | F1 ↑ | ACC ↑ | F1 ↑ | ACC ↑ | F1 ↑ | ACC ↑ | F1 ↑ |
| Gemini-2.5-pro | **0.52** | **0.81** | **0.33** | **0.81** | **0.20** | **0.78** | **0.35** | **0.80** |
| GPT-4o | 0.21 | 0.55 | 0.07 | 0.59 | 0.06 | 0.57 | 0.11 | 0.57 |
| Claude-3.7-sonnet | 0.29 | 0.66 | 0.11 | 0.62 | 0.08 | 0.62 | 0.16 | 0.64 |
| Grok-3 | 0.26 | 0.59 | 0.17 | 0.63 | 0.03 | 0.56 | 0.15 | 0.59 |
| Doubao-1.5-vision-pro | 0.26 | 0.63 | 0.04 | 0.61 | 0.03 | 0.54 | 0.11 | 0.60 |
| Kimi-VL-A3B | 0.03 | 0.29 | 0.00 | 0.00 | 0.00 | 0.00 | 0.00 | 0.28 |
| Qwen2.5-VL-72B | 0.06 | 0.44 | 0.00 | 0.49 | 0.00 | 0.43 | 0.02 | 0.45 |

Table 13: Circuit Partition Task Performance Across Different Models

| Models | Reasoning Task (ACC ↑) | | | |
|---|---|---|---|---|
| | Op-amp | LDO | Bandgap | Comparator |
| Gemini-2.5-pro | 0.94 | 0.90 | 0.88 | **0.96** |
| GPT-4o | 0.78 | 0.84 | 0.64 | 0.84 |
| Claude-3.7-sonnet | **0.95** | **0.96** | 0.78 | **0.96** |
| Grok-3 | 0.60 | 0.72 | 0.52 | 0.60 |
| Doubao-1.5-vision-pro | 0.90 | 0.84 | 0.80 | 0.80 |
| Kimi-VL-A3B | 0.81 | 0.46 | **0.86** | 0.84 |
| Qwen2.5-VL-72B | 0.81 | 0.90 | 0.72 | 0.84 |

Table 14: Reasoning Task Performance Across Different Models

| Models | Caption Generation Task(ACC ↑) | | |
|---|---|---|---|
| | **Undergraduate** | **Graduate** | **Total** |
| Gemini-2.5-pro | 0.63 | 0.72 | 0.70 |
| GPT-4o | 0.58 | 0.63 | 0.61 |
| Claude-3.7-sonnet | **0.89** | **1.00** | **0.98** |
| Grok-3 | 0.21 | 0.47 | 0.41 |
| Doubao-1.5-vision-pro | 0.74 | 0.69 | 0.70 |
| Kimi-VL-A3B | 0.74 | 0.70 | 0.71 |
| Qwen2.5-VL-72B | **0.89** | 0.75 | 0.78 |

Table 15: Caption Generation Task Performance Across Different Models

| Models | TQA Task(ACC ↑) | | | |
|---|---|---|---|---|
| | **Undergraduate** | **Graduate** | **Engineer** | **Total** |
| Gemini-2.5-pro | 0.89 | 0.89 | 0.39 | 0.72 |
| GPT-4o | 0.88 | 0.88 | **0.58** | **0.78** |
| Claude-3.7-sonnet | **0.90** | 0.88 | 0.45 | 0.74 |
| Grok-3 | 0.86 | 0.84 | 0.53 | 0.74 |
| Doubao-1.5-vision-pro | 0.88 | 0.88 | 0.52 | 0.76 |
| Kimi-VL-A3B | 0.80 | 0.82 | 0.16 | 0.59 |
| Qwen2.5-VL-72B | 0.85 | 0.85 | 0.37 | 0.69 |
| DeepSeek-R1 | 0.89 | **0.90** | **0.58** | 0.77 |

Table 16: TQA Task Performance Across Different Models (Decimal Format)

## B.3 RESULTS OF DESIGN TASKS

### B.3.1 CIRCUIT DESIGN TASK

| Id | Type | Circuit Description | Id | Type | Circuit Description |
|----|------|---------------------|----|------|---------------------|
| 1 | Amplifier | Single-stage common-source amp. with R load | 15 | Op-amp | Telescopic cascode op-amp |
| 2 | Amplifier | 3-stage common-source amp. with R load | 16 | Oscillator | RC phase-shift oscillator |
| 3 | Amplifier | Common-drain amp. with R load | 17 | Oscillator | A Wien bridge oscillator |
| 4 | Amplifier | Single-stage common-gate amp. with R load | 18 | Integrator | Op-amp integrator |
| 5 | Amplifier | Single-stage cascode amp. with R load | 19 | Differentiator | Op-amp differentiator |
| 6 | Inverter | NMOS inverter with R load | 20 | Adder | Op-amp adder |
| 7 | Inverter | Logical inverter with NMOS and PMOS | 21 | Subtractor | Op-amp subtractor |
| 8 | Current mirror | NMOS constant current source with R load | 22 | Schmitt trigger | Non-inverting Schmitt trigger |
| 9 | Amplifier | 2-stage amp. with Miller compensation | 23 | VCO | Voltage-controlled oscillator |
| 10 | Amplifier | Common-source amp. with diode-connected load | 24 | Bandgap | A classic brokaw bandgap reference |
| 11 | Op-amp | Differential op-amp with current mirror load | 25 | Comparator | A low offset voltage dual comparator |
| 12 | Current mirror | Cascode current mirror | 26 | LDO | 1A low dropout voltage regulator |
| 13 | Op-amp | Single-stage common-source op-amp with R loads | 27 | PLL | Phase-locked loop |
| 14 | Op-amp | 2-stage differential op-amp with active loads | 28 | SAR-ADC | Successive approximation register ADC |

Table 17: Circuit block library (two parallel lists), with column-specific background coloring(blue means simple, green means complex, red means system level

| Models | 1. Amplifier | | | 2. Amplifier | | | 3. Amplifier | | | 4. Amplifier | | |
|--------|------|------|-------|------|------|-------|------|------|-------|------|------|-------|
| | p@3 | p@5 | p@10 | p@3 | p@5 | p@10 | p@3 | p@5 | p@10 | p@3 | p@5 | p@10 |
| Gemini-2.5-pro | 1 | 1 | 1 | 1 | 1 | 1 | 0.67 | 0.67 | 0.67 | 1 | 1 | 1 |
| Gpt-4o | 1 | 1 | 1 | 1 | 1 | 1 | 0 | 1 | 1 | 0.33 | 0.33 | 0.33 |
| Claude-3.7-sonnet | 1 | 1 | 1 | 1 | 1 | 1 | 0.33 | 0.67 | 1 | 0.67 | 0.67 | 0.67 |
| Grok-3 | 1 | 1 | 1 | 1 | 1 | 1 | 0 | 1 | 1 | 0.67 | 0.67 | 0.67 |
| Doubao-1.5-vision-pro | 0.67 | 0.67 | 0.67 | 0.43 | 0.43 | 0.43 | 0.33 | 0.33 | 0.33 | 0.13 | 0.13 | 0.13 |
| Kimi-VL-A3B | 0 | 0 | 0 | 0 | 0 | 0 | 0 | 0 | 0 | 0 | 0 | 0 |
| Qwen2.5-VL-72B | 0 | 0 | 0 | 0 | 0 | 0 | 0 | 0 | 0 | 0 | 0 | 0 |

Table 18: Per-circuit pass@k scores (k = 3, 5, 10) for CKT1–CKT4 across multiple models

| Models | 5. Amplifier | | | 6. Inverter | | | 7. Inverter | | | 8. Current mirror | | |
|--------|------|------|-------|------|------|-------|------|------|-------|------|------|-------|
| | p@3 | p@5 | p@10 | p@3 | p@5 | p@10 | p@3 | p@5 | p@10 | p@3 | p@5 | p@10 |
| Gemini-2.5-pro | 1 | 1 | 1 | 1 | 1 | 1 | 1 | 1 | 1 | 1 | 1 | 1 |
| GPT-4o | 1 | 1 | 1 | 1 | 1 | 1 | 1 | 1 | 1 | 1 | 1 | 1 |
| Claude-3.7-sonneet | 1 | 1 | 1 | 1 | 1 | 1 | 1 | 1 | 1 | 1 | 1 | 1 |
| Grok-3 | 1 | 1 | 1 | 1 | 1 | 1 | 1 | 1 | 1 | 1 | 1 | 1 |
| Doubao-1.5-vision-pro | 0.23 | 0.23 | 0.23 | 0.23 | 0.23 | 0.23 | 0.49 | 0.49 | 0.49 | 1 | 1 | 1 |
| Kimi-VL-A3B | 0 | 0 | 0 | 0 | 0 | 0 | 0 | 0 | 0 | 0 | 0 | 0 |
| Qwen2.5-VL-72B | 0 | 0 | 0 | 0 | 0 | 0 | 0 | 0 | 0 | 0 | 0 | 0 |

Table 19: Per-circuit pass@k scores (k = 3, 5, 10) for CKT5–CKT8 across multiple models

| Model | 9. Amplifier | | | 10. Amplifier | | | 11. Op-amp | | | 12. Current mirror | | |
|---|---|---|---|---|---|---|---|---|---|---|---|---|
| | p@3 | p@5 | p@10 | p@3 | p@5 | p@10 | p@3 | p@5 | p@10 | p@3 | p@5 | p@10 |
| Gemini-2.5-pro | 1 | 1 | 1 | 0.61 | 0.61 | 0.61 | 0.48 | 0.48 | 0.48 | 0.24 | 0.24 | 0.24 |
| GPT-4o | 1 | 1 | 1 | 0.43 | 0.43 | 0.43 | 0.22 | 0.22 | 0.22 | 0 | 0 | 0 |
| Claude-3.7-sonnet | 1 | 1 | 1 | 0.87 | 0.87 | 0.87 | 0.65 | 0.65 | 0.65 | 0.62 | 0.62 | 0.62 |
| Grok-3 | 1 | 1 | 1 | 0.86 | 0.86 | 0.86 | 0.29 | 0.29 | 0.29 | 0.33 | 0.33 | 0.33 |
| Doubao-1.5-vision-pro | 0.27 | 0.27 | 0.27 | 0.61 | 0.61 | 0.61 | 0.33 | 0.33 | 0.33 | 0.31 | 0.31 | 0.31 |
| Kimi-VL-A3B | 0 | 0 | 0 | 0 | 0 | 0 | 0 | 0 | 0 | 0 | 0 | 0 |
| Qwen2.5-VL-72B | 0 | 0 | 0 | 0 | 0 | 0 | 0 | 0 | 0 | 0 | 0 | 0 |

Table 20: Per-circuit pass@k scores (k = 3, 5, 10) for CKT9–CKT12 across multiple models

| Model | 13. Op-amp | | | 14. Op-amp | | | 15. Op-amp | | | 16. Oscillator | | |
|---|---|---|---|---|---|---|---|---|---|---|---|---|
| | p@3 | p@5 | p@10 | p@3 | p@5 | p@10 | p@3 | p@5 | p@10 | p@3 | p@5 | p@10 |
| Gemini-2.5-pro | 0.43 | 0.43 | 0.43 | 1 | 1 | 1 | 0.33 | 0.4 | 0.1 | 0 | 0 | 0 |
| GPT-4o | 0.32 | 0.32 | 0.32 | 0.21 | 0.21 | 0.21 | 0 | 0.2 | 0.1 | 0.33 | 0.6 | 0.3 |
| Claude-3.7-sonnet | 1 | 1 | 1 | 0.71 | 0.71 | 0.71 | 1 | 1 | 0.9 | 0 | 0 | 0 |
| Grok-3 | 0.69 | 0.69 | 0.69 | 0.89 | 0.89 | 0.89 | 1 | 1 | 1 | 0 | 0 | 0 |
| Doubao-1.5-vision-pro | 0.68 | 0.68 | 0.68 | 0 | 0 | 0 | 0.33 | 0.2 | 0.1 | 0.33 | 0.2 | 0.1 |
| Kimi-VL-A3B | 0 | 0 | 0 | 0 | 0 | 0 | 0 | 0 | 0 | 0 | 0 | 0 |
| Qwen2.5-VL-72B | 0 | 0 | 0 | 0 | 0 | 0 | 0 | 0 | 0 | 0 | 0 | 0 |

Table 21: Per-circuit pass@k scores (k = 3, 5, 10) for CKT13–CKT16 across multiple models

| Model | 17. Oscillator | | | 18. Integrator | | | 19. Differentiator | | | 20. Adder | | |
|---|---|---|---|---|---|---|---|---|---|---|---|---|
| | p@3 | p@5 | p@10 | p@3 | p@5 | p@10 | p@3 | p@5 | p@10 | p@3 | p@5 | p@10 |
| Gemini-2.5-pro | 0 | 0 | 0 | 0.33 | 0.4 | 0.1 | 0.2 | 0.1 | 0.33 | 0.4 | 0.1 | 0.33 |
| GPT-4o | 1 | 0.2 | 0.1 | 0.33 | 0.6 | 0.1 | 0.4 | 0.1 | 0 | 0.4 | 0.5 | 0.33 |
| Claude-3.7-sonnet | 0.33 | 1 | 0.3 | 0.33 | 0.2 | 0.1 | 0.33 | 0.2 | 0.1 | 1 | 0.6 | 0.1 |
| Grok-3 | 0.33 | 0.6 | 0.3 | 0.67 | 0 | 0.3 | 0.67 | 0.8 | 0.1 | 0.33 | 0.4 | 0.2 |
| Doubao-1.5-vision-pro | 0.33 | 0.2 | 0.1 | 1 | 0.2 | 0.4 | 0.67 | 0.2 | 0.1 | 0.33 | 0.2 | 0.1 |
| Kimi-VL-A3B | 0 | 0 | 0 | 0 | 0 | 0 | 0 | 0 | 0 | 0 | 0 | 0 |
| Qwen2.5-VL-72B | 0 | 0 | 0 | 0 | 0 | 0 | 0 | 0 | 0 | 0 | 0 | 0 |

Table 22: Per-circuit pass@k scores (k = 3, 5, 10) for CKT17–CKT20 across multiple models

| Model | 21. Subtractor | | | 22. Schmitt trigger | | | 23. VCO | | | 24. Bandgap | | |
|---|---|---|---|---|---|---|---|---|---|---|---|---|
| | p@3 | p@5 | p@10 | p@3 | p@5 | p@10 | p@3 | p@5 | p@10 | p@3 | p@5 | p@10 |
| Gemini-2.5-pro | 0.4 | 0.1 | 0.33 | 0.33 | 0.4 | 0.1 | 0.67 | 1 | 0.9 | 0 | 0 | 0 |
| GPT-4o | 0.6 | 0.2 | 0.33 | 0.2 | 0.33 | 0.4 | 0.5 | 0.33 | 0.6 | 0 | 0 | 0 |
| Claude-3.7-sonnet | 1 | 0.8 | 0.2 | 0.33 | 0.4 | 0.1 | 0.33 | 0.2 | 0.2 | 0 | 0 | 0 |
| Grok-3 | 0.33 | 0.2 | 1 | 0.33 | 0.2 | 0.1 | 0.6 | 0.5 | 0.33 | 0 | 0 | 0 |
| Doubao-1.5-vision-pro | 0.33 | 0.2 | 0.1 | 0.33 | 0.2 | 0.1 | 0.67 | 0.2 | 0.1 | 0 | 0 | 0 |
| Kimi-VL-A3B | 0 | 0 | 0 | 0 | 0 | 0 | 0 | 0 | 0 | 0 | 0 | 0 |
| Qwen2.5-VL-72B | 0 | 0 | 0 | 0 | 0 | 0 | 0 | 0 | 0 | 0 | 0 | 0 |

Table 23: Per-circuit pass@k scores (k = 3, 5, 10) for CKT21–CKT24 across multiple models

| Model | 25. Comparator | | | 26. LDO | | | 27. PLL | | | 28. SAR-ADC | | |
|---|---|---|---|---|---|---|---|---|---|---|---|---|
| | p@3 | p@5 | p@10 | p@3 | p@5 | p@10 | p@3 | p@5 | p@10 | p@3 | p@5 | p@10 |
| Gemini-2.5-pro | 0 | 0 | 0 | 0 | 0 | 0 | 0 | 0 | 0 | 0 | 0 | 0 |
| GPT-4o | 0 | 0 | 0 | 0 | 0 | 0 | 0 | 0 | 0 | 0 | 0 | 0 |
| Claude-3.7-sonnet | 0 | 0 | 0 | 0 | 0 | 0 | 0 | 0 | 0 | 0 | 0 | 0 |
| Grok-3 | 0 | 0 | 0 | 0 | 0 | 0 | 0 | 0 | 0 | 0 | 0 | 0 |
| Doubao-1.5-vision-pro | 0 | 0 | 0 | 0 | 0 | 0 | 0 | 0 | 0 | 0 | 0 | 0 |
| Kimi-VL-A3B | 0 | 0 | 0 | 0 | 0 | 0 | 0 | 0 | 0 | 0 | 0 | 0 |
| Qwen2.5-VL-72B | 0 | 0 | 0 | 0 | 0 | 0 | 0 | 0 | 0 | 0 | 0 | 0 |

Table 24: Per-circuit pass@k scores (k = 3, 5, 10) for CKT25–CKT28 across multiple models

### B.3.2 TESTBENCH DESIGN TASK

| ID | Circuit Type | # Metrics | ID | Circuit Type | # Metrics |
|---|---|---|---|---|---|
| 1 | Cross-coupled differential amplifier | CMRR, DC gain, GBW, Phase margin, Power, PSR, SR | 7 | LDO | LDR, LNR, Drop voltage, DC gain, Phase margin, PSR, Offset |
| 2 | Comparator | Delay, Offset | 8 | VCO | Jitter, Phase noise |
| 3 | Bootstrap | ENOB | 9 | Unit capacitor | MC-mismatch |
| 4 | Telescopic cascode OTA | CMRR, DC gain, GBW, Phase margin, Power, PSR, SR | 10 | Folded cascode OTA | DC gain, SR, Phase margin, GBW, Power |
| 5 | PLL | Jitter, Phase noise | 11 | SAR-ADC | ENOB |
| 6 | MOS_Ron | Ron | 12 | Bandgap | BuildingupV, Noise, PSR, TC |

Table 25: Testbench design tasks with metrics to be simulated.

| Models | 1. Cross-coupled differential amplifier | | | | | | | 2. Comparator | | 3. Boot-strap |
|---|---|---|---|---|---|---|---|---|---|---|
| | CMRR | DC gain | GBW | Phase margin | Power | PSR | SR | Delay | Offset | ENOB |
| Gemini-2.5-pro | 0 | 0 | 0 | 0 | 0 | 0 | 0 | 0 | 0 | 0 |
| GPT-4o | **0 (1)** | 0 | 0 | 0 | **0(4)** | **0(2)** | 0 | **0(2)** | **0(2)** | 0 |
| Claude-3.7-sonnet | 0 | 0 | 0 | 0 | 0 | 0 | 0 | 0 | 0 | 0 |
| Grok-3 | 0 | 0 | 0 | 0 | 0 | 0 | 0 | 0 | 0 | 0 |
| Doubao-1.5-vision-pro | 0 | 0 | 0 | 0 | 0 | 0 | 0 | 0 | 0 | 0 |
| Kimi-VL-A3B | 0 | 0 | 0 | 0 | 0 | 0 | 0 | 0 | 0 | 0 |
| Qwen2.5-VL-72B | 0 | 0 | 0 | 0 | 0 | 0 | 0 | 0 | 0 | 0 |

Table 26: Cross-coupled differential amplifier, Comparator, and Bootstrap Circuit Design Performance Across Different Models

| Models | 4. Telescopic cascode OTA | | | | | | | 5. PLL | | 6. MOS_Ron |
|---|---|---|---|---|---|---|---|---|---|---|
| | CMRR | DC gain | GBW | Phase margin | Power | PSR | SR | Jitter | Phase noise | Ron |
| Gemini-2.5-pro | 0 | 0 | 0 | 0 | 0 | 0 | 0 | 0 | 0 | 0 |
| GPT-4o | 0 | 0 | 0 | 0 | 0 | 0 | 0 | 0 | 0 | 0 |
| Claude-3.7-sonnet | 0 | 0 | 0 | 0 | 0 | 0 | 0 | 0 | 0 | 0 |
| Grok-3 | 0 | 0 | 0 | 0 | 0 | 0 | 0 | 0 | 0 | 0 |
| Doubao-1.5-vision-pro | 0 | 0 | 0 | 0 | 0 | 0 | 0 | 0 | 0 | 0 |
| Kimi-VL-A3B | 0 | 0 | 0 | 0 | 0 | 0 | 0 | 0 | 0 | 0 |
| Qwen2.5-VL-72B | 0 | 0 | 0 | 0 | 0 | 0 | 0 | 0 | 0 | 0 |

Table 27: Telescopic cascode OTA, PLL, and MOS_Ron Performance Across Different Models

| Models | 7. LDO | | | | | | | 8. VCO | | 9. Unit capacitor |
|---|---|---|---|---|---|---|---|---|---|---|
| | LDR | LNR | Drop voltage | DC gain | Phase margin | PSR | Offset | Jitter | Phase noise | MC-mismatch |
| Gemini-2.5-pro | 0 | 0 | 0 | 0 | 0 | 0 | 0 | 0 | 0 | 0 |
| GPT-4o | 0 | 0 | 0 | 0 | 0 | 0 | 0 | 0 | 0 | **0(3)** |
| Claude-3.7-sonnet | 0 | 0 | 0 | 0 | 0 | 0 | 0 | 0 | 0 | 0 |
| Grok-3 | 0 | 0 | 0 | 0 | 0 | 0 | 0 | 0 | 0 | 0 |
| Doubao-1.5-vision-pro | 0 | 0 | 0 | 0 | 0 | 0 | 0 | 0 | 0 | 0 |
| Kimi-VL-A3B | 0 | 0 | 0 | 0 | 0 | 0 | 0 | 0 | 0 | 0 |
| Qwen2.5-VL-72B | 0 | 0 | 0 | 0 | 0 | 0 | 0 | 0 | 0 | 0 |

Table 28: LDO, VCO, and Unit Capacitor Performance Across Different Models

| Models | 10. Folded cascode OTA | | | | | 11. SAR-ADC | 12. Bandgap | | | |
|---|---|---|---|---|---|---|---|---|---|---|
| | DC gain | SR | Phase margin | GBW | Power | ENOB | BuildiupV | Noise | PSR | TC |
| Gemini-2.5-pro | 0 | 0 | 0 | 0 | 0 | 0 | 0 | 0 | 0 | 0 |
| GPT-4o | 0 | 0 | 0 | 0 | 0 | 0 | 0 | **0(1)** | 0 | 0 |
| Claude-3.7-sonnet | 0 | 0 | 0 | 0 | 0 | 0 | 0 | 0 | 0 | 0 |
| Grok-3 | 0 | 0 | 0 | 0 | 0 | 0 | 0 | 0 | 0 | 0 |
| Doubao-1.5-vision-pro | 0 | 0 | 0 | 0 | 0 | 0 | 0 | 0 | 0 | 0 |
| Kimi-VL-A3B | 0 | 0 | 0 | 0 | 0 | 0 | 0 | 0 | 0 | 0 |
| Qwen2.5-VL-72B | 0 | 0 | 0 | 0 | 0 | 0 | 0 | 0 | 0 | 0 |

Table 29: Folded Cascode OTA, SAR-ADC, and Bandgap Performance Across Different Models

## C   TEST EXAMPLES

### C.1   EXAMPLES OF PERCEPTION TASKS

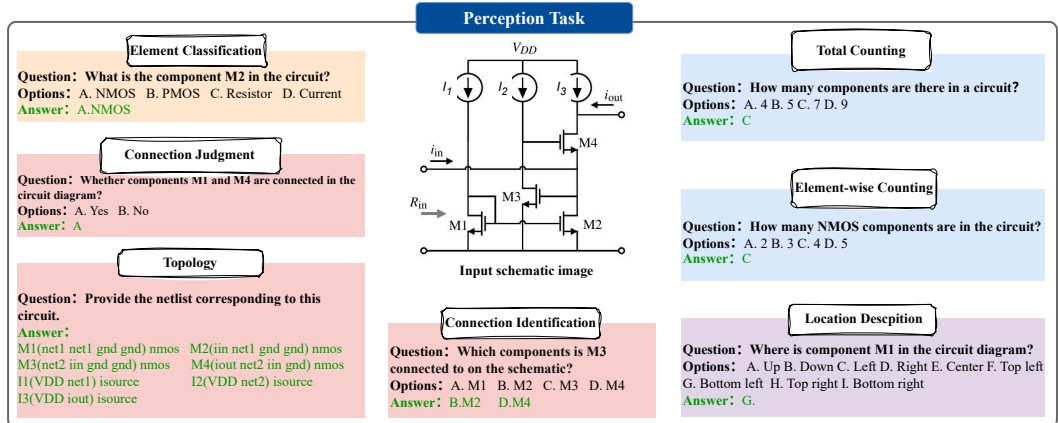

Figure 7: Example of **Perception** task in AMSbench

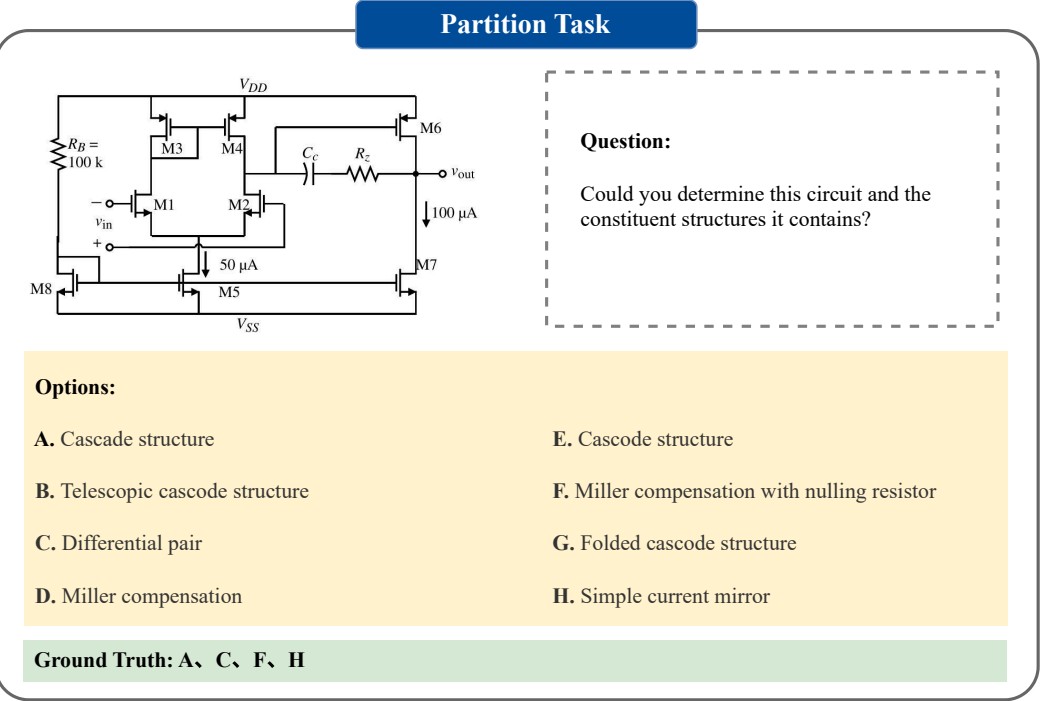

Figure 8: Example of **Partition** task in AMSbench

## C.2 EXAMPLES OF ANALYSIS TASKS

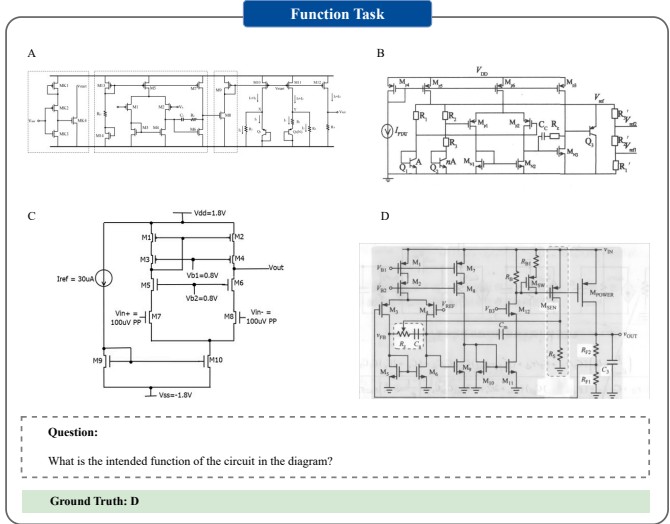

**Reasoning Task**

**Question:**

According to the following circuit diagram，tell me why the circuit represented by this diagram is an operational amplifier?

**Options:**

**A.** Through the coordination of the differential input stage, the high-gain intermediate stage and the output stage, high voltage gain and differential signal processing capabilities are achieved, which conforms to the core characteristics of an operational amplifier, namely "high gain, differential input, and signal amplification ability". Therefore, it belongs to the operational amplifier circuit.

**B.** This circuit does not meet the typical structural requirements of an operational amplifier: An operational amplifier usually consists of an input stage (differential amplification), a high-gain intermediate stage, and an output stage (such as a push-pull structure), and it needs to have characteristics such as high gain and frequency compensation. However, this circuit only exhibits a differential input structure (M1, M2), lacking the crucial intermediate high-gain amplification stage and the standard output stage (such as a push-pull output circuit), and it also does not have components related to frequency compensation. As a result, it cannot achieve the core functions required for an operational amplifier, such as high gain, low output impedance, and stable amplification. Therefore, it does not belong to an operational amplifier circuit.

**C.** An operational amplifier needs to have a complete frequency compensation mechanism to ensure the stability of the circuit (for example, by implementing phase margin adjustment through a Miller capacitor, etc.). However, this circuit has not designed any structure related to frequency compensation. It cannot effectively suppress self-excited oscillation under different input signal frequencies, nor can it ensure the stability during closed-loop operation. This is a clear deficiency compared to the stable amplification characteristics (including the stability of the frequency response) that an operational amplifier must meet. Therefore, it does not belong to an operational amplifier circuit.

**D.** The circuit diagram does not represent an operational amplifier because it lacks the key characteristics of an operational amplifier, such as the feedback mechanism, multi-stage amplification, and buffered output structures. Its output stage is simple and cannot achieve the high gain and stable performance required by an operational amplifier.

**Ground Truth: A**

Figure 9: Examples of **Reasoning** task in AMSbench

**Function Task**

A

B

C

D

**Question:**

What is the intended function of the circuit in the diagram?

**Ground Truth: D**

Figure 10: Example of **Function(image as options)** task in AMSbench

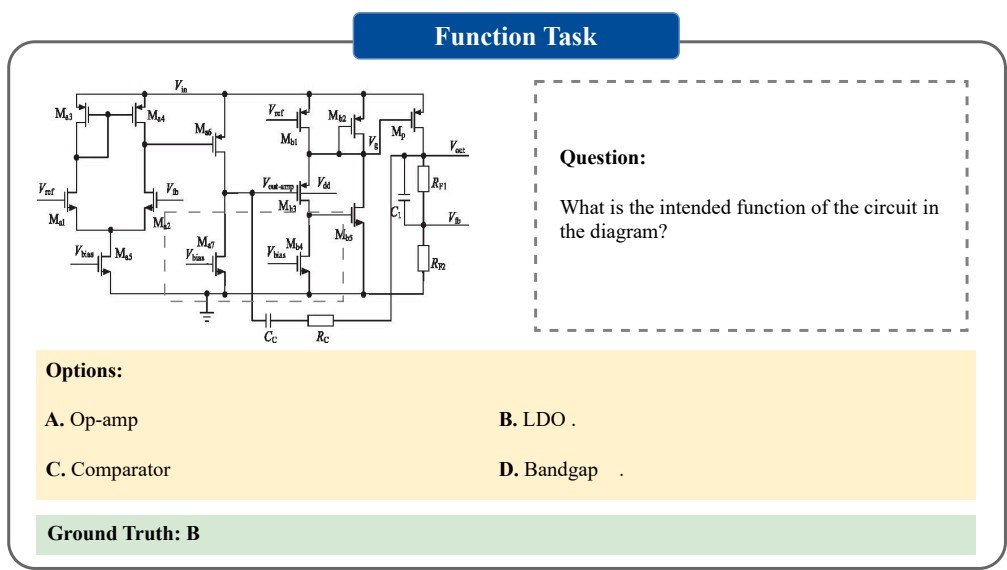

**Figure 11: Example of Function(text as options) task in AMSbench**

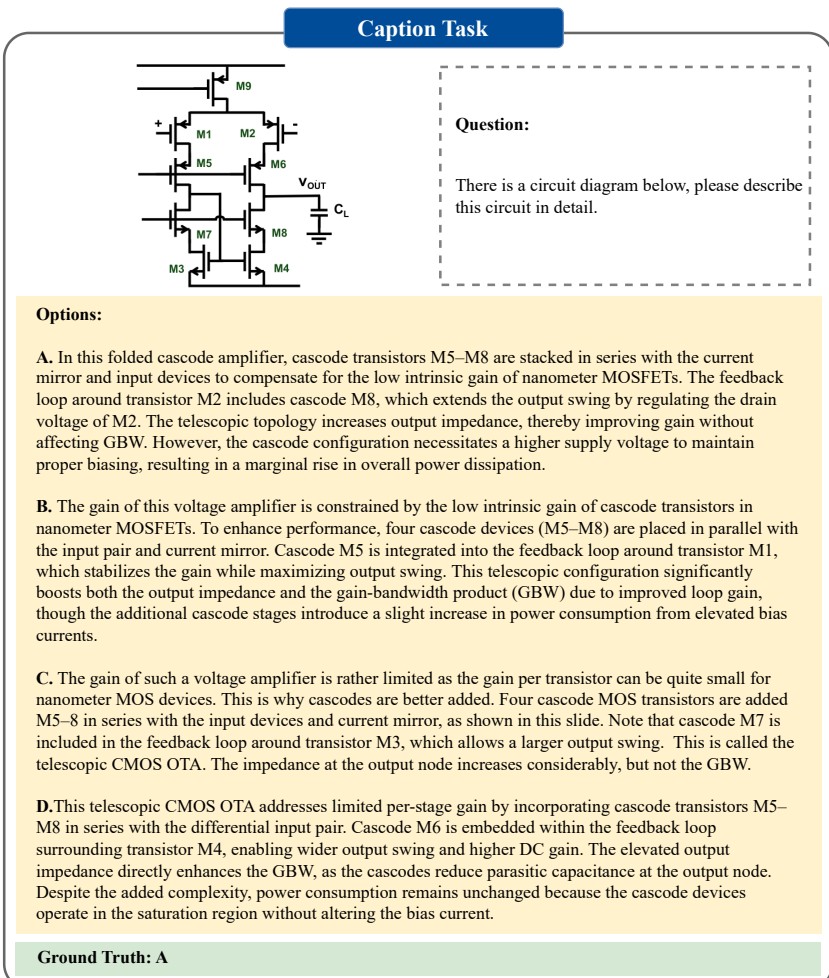

**Figure 12: Example of Caption task in AMSbench**

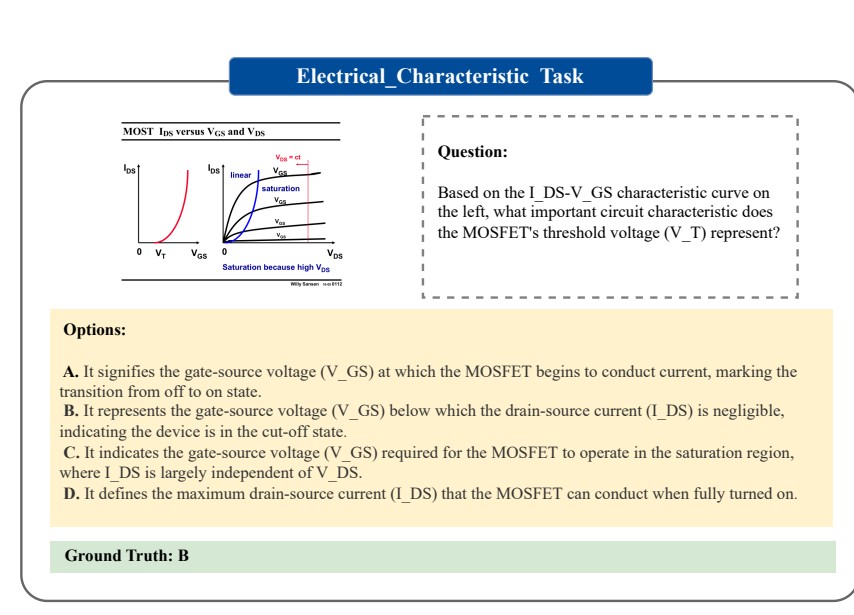

Figure 13: Example of **Electrical Characteristic** task in AMSbench

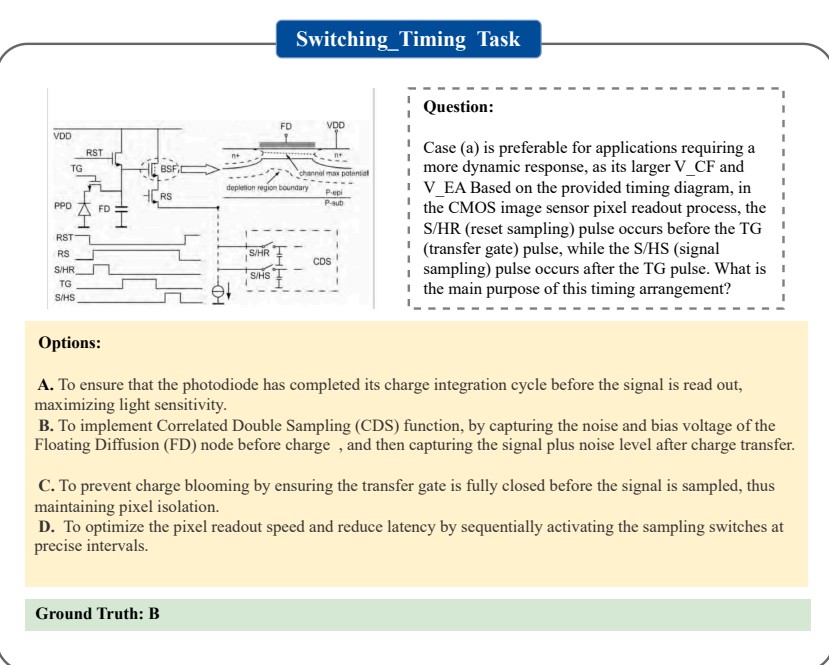

Figure 14: Example of **Switching Timing** task in AMSbench

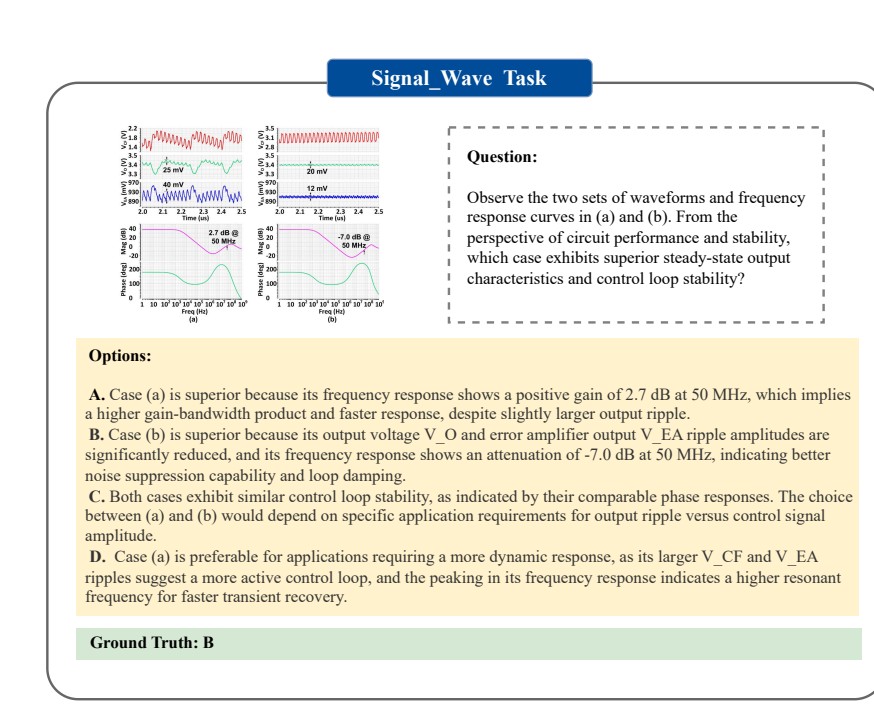

Figure 15: Example of **Signal Wave** task in AMSbench

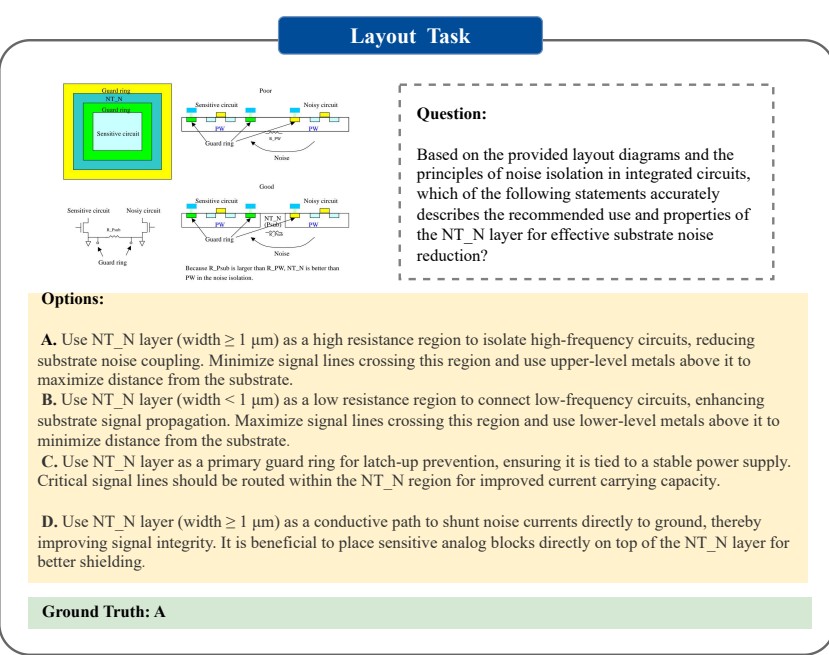

Figure 16: Example of **Layout** task in AMSbench

## C.3 EXAMPLES OF DESIGN TASKS

**User**

You are an analog integrated circuits expert.

**Assistant (Circuit Design Prompt)**

You aim to design a topology for a given circuit described in the text.
Please ensure your designed circuit topology works properly and achieves the design requirements.

Here is an example:

**Question:**
Design a 2-stage amplifier (first stage: a common-source stage with current-source load, second stage: a common-source stage with resistor load).

Input node name: Vin, Vbias.

Output node name: Vout.

**Answer:**

**Task 1**

**Components Needed**
- **NMOS Transistors**: M1 and M3
- **PMOS Transistors**: M2 (used as the current source in the first stage)
- **Resistors**: R1 for the second stage load
- **Power Supply**: Vdd for DC supply
- **Input Signal Source**: Vin, Vbias for biasing and signal input
- **Capacitors**: Not specified but can be included for coupling and bypass applications if required

**Stage 1: Common-Source Amplifier with Current Source Load**
1. **Transistor Setup**:
   - **M1**(NMOS) as the main amplifying transistor.
   - Gate of **M1** is connected to the input node **Vin**.
   - Source of **M1** connected to the ground.
   - Drain of **M1** connected to the drain of **M2**.

2. **Biasing**:
   - **Vin** provides the input signal.
   - **Vbias** is used to bias **M2** (PMOS), ensuring it operates as a current source.

3. **Current Source Load (M2)**:
   - **M2**, a PMOS transistor, is configured as a current source.
   - The source of **M2** is connected to **Vdd**, and its gate is connected to **Vbias**.
   - Drain of **M2** is connected to the drain of **\*\*M1\*\***, providing a high-impedance load.

**Stage 2: Common-Source Amplifier with Resistor Load**
1. **Transistor Setup**:
   - **M3** (NMOS) as the main amplifying transistor for the second stage.
   - Gate of **M3** connected to the drain of **M1**.
   - Source of **M3** connected to the ground.
   - Drain of **M3** connected to **Vout** through resistor **R1**.

2. **Load and Coupling**:
   - **R1** connects the drain of **M3** to **Vdd**. This resistor converts the current through **M3** into an output voltage.

**Assistant (Circuit Design Prompt)**

**Task 2**
```
from PySpice.Spice.Netlist import Circuit
from PySpice.Unit import *

circuit = Circuit('Two-Stage Amplifier')
# Define the MOSFET models
circuit.model('nmos_model', 'nmos', level=1, kp=100e-6, vto=0.5)
circuit.model('pmos_model', 'pmos', level=1, kp=50e-6, vto=-0.5)

# Power Supplies for the power and input signal

circuit.V('dd', 'Vdd', circuit.gnd, 5.0) # 5V power supply
circuit.V('in', 'Vin', circuit.gnd, 1.0) # 1V input for bias voltage
circuit.V('bias', 'Vbias', circuit.gnd, 4.0) # 4V input for bias voltage

# First Stage: Common-Source with Active Load
# parameters: name, drain, gate, source, bulk, model, w, l
circuit.MOSFET('1', 'Drain1', 'Vin', circuit.gnd, circuit.gnd,
model='nmos_model', w=50e-6, l=1e-6)
circuit.MOSFET('2', 'Drain1', 'Vbias', 'Vdd', 'Vdd', model='pmos_model',
w=100e-6, l=1e-6)

# Second Stage: Common-Source with Resistor Load
circuit.MOSFET('3', 'Vout', 'Drain1', circuit.gnd, circuit.gnd,
model='nmos_model', w=100e-6, l=1e-6)
circuit.R('1', 'Vout', 'Vdd', 1@u_kΩ)

# Analysis Part
simulator = circuit.simulator()
```

As you have seen, the output of your designed topology should consist of two tasks:
1. Give a detailed design plan about all devices and their interconnectivity nodes and properties.
2. Write a complete Python code, describing the topology of integrated analog circuits according to the design plan.

Please make sure your Python code is compatible with PySpice.
Please give the runnable code without any placeholders.
Do not write other redundant codes after 'simulator = circuit.simulator()'.

There are some tips you should remember all the time:
   1. For the MOSFET definition circuit.MOSFET(name, drain, gate, source, bulk, model, w=w1,l=l1), be careful about the parameter sequence.
   2. You should connect the bulk of a MOSFET to its source.
   3. Please use the MOSFET threshold voltage, when setting the bias voltage.
   4. Avoid giving any AC voltage in the sources, just consider the operating points.
   5. Make sure the input and output node names appear in the circuit.
   6. Avoid using subcircuits.
   7. Use nominal transistor sizing.
   8. Assume the Vdd = 5.0 V.

**Question:**
Design [TASK].
Input node name: [INPUT].
Output node name: [OUTPUT].

**Answer: (......)**

Figure 17: Prompt of **Circuit Design** task in AMSbench

1458
1459
1460
1461
1462
1463
1464
1465
1466
1467
1468
1469
1470
1471
1472
1473
1474
1475
1476
1477
1478
1479
1480
1481
1482
1483
1484
1485
1486
1487
1488
1489
1490
1491
1492
1493
1494
1495
1496
1497
1498
1499
1500
1501
1502
1503
1504
1505
1506
1507
1508
1509
1510
1511

**User**

You are an analog integrated circuits expert.

**Assistant（Testbench Design Prompt）**

You are an analog circuit design engineer with over 10 years of experience, familiar with the Cadence Spectre simulation environment, CMOS circuit modeling, and various circuit performance testing methods.

I will provide a netlist file and schematic diagram of a "certain circuit" (given in both text and image formats). Based on the netlist file, please write a complete Spectre simulation input file (.cir format) to test the "certain performance" of the circuit, while keeping the core structure of the circuit unchanged.

**Requirements:**

Add the necessary excitation sources (such as voltage sources, current sources, etc.) and appropriate simulation analysis statements (such as ac, dc, tran, noise, etc.) to test the target performance.

The generated .cir file should be directly runnable in Spectre without any additional modifications.

Briefly explain your test configuration approach (including input signals, biasing methods, load, power supply, etc.).

**Please output exactly five separate Spectre testbench code blocks. Each block must be wrapped in triple backticks, like:**
// testbench #1
...
// testbench #2
...

**The example of the writing format of the testbench is referred to as follows:**
<testbench_format.cir>

Figure 18: Prompt of **Testbench Design** task in AMSbench

# D CASE STUDY

## D.1 PERCEPTION TASK FOR ERROR ANALYSIS

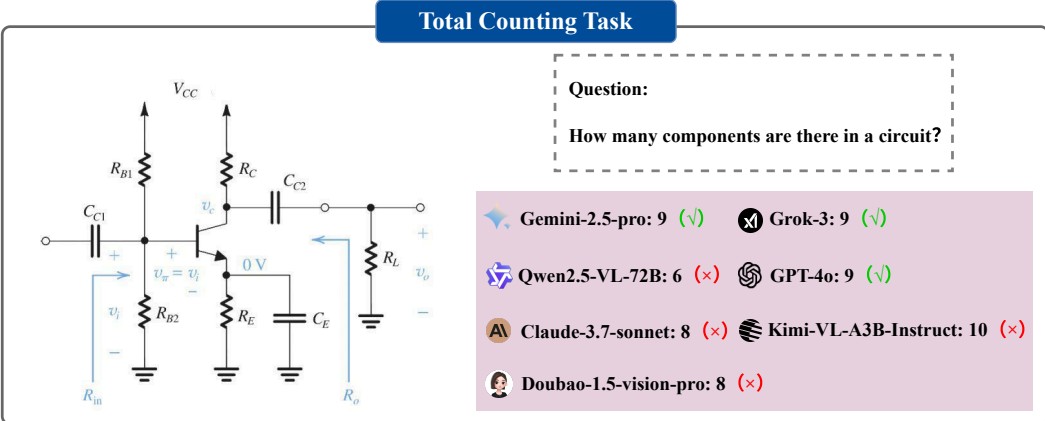

Figure 19: Example of **Total Counting** task across models

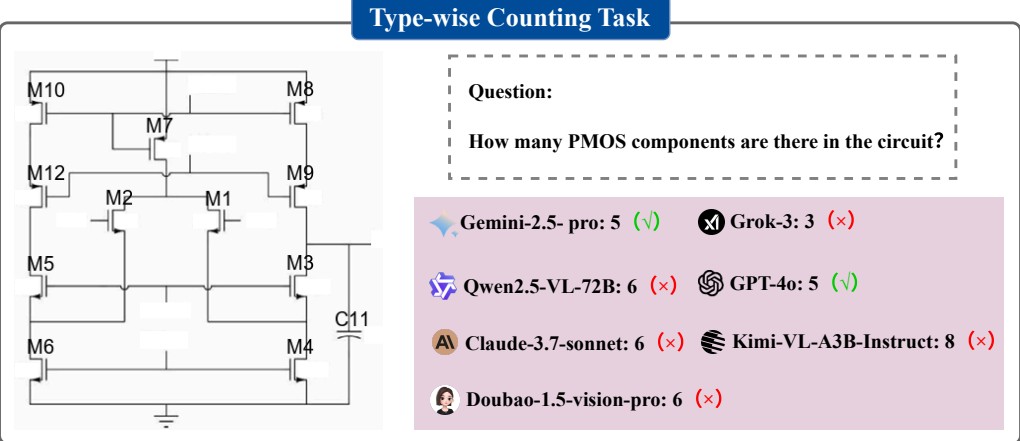

Figure 20: Example of **Type-wise Counting** task across models

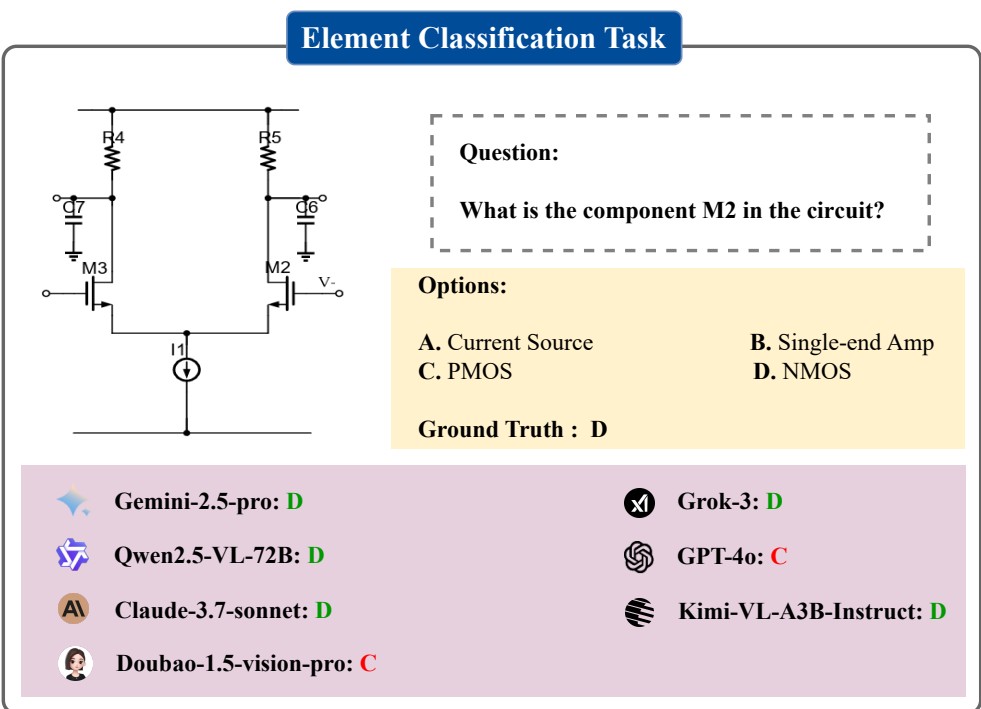

Figure 21: Example of **Element Classification** task across models

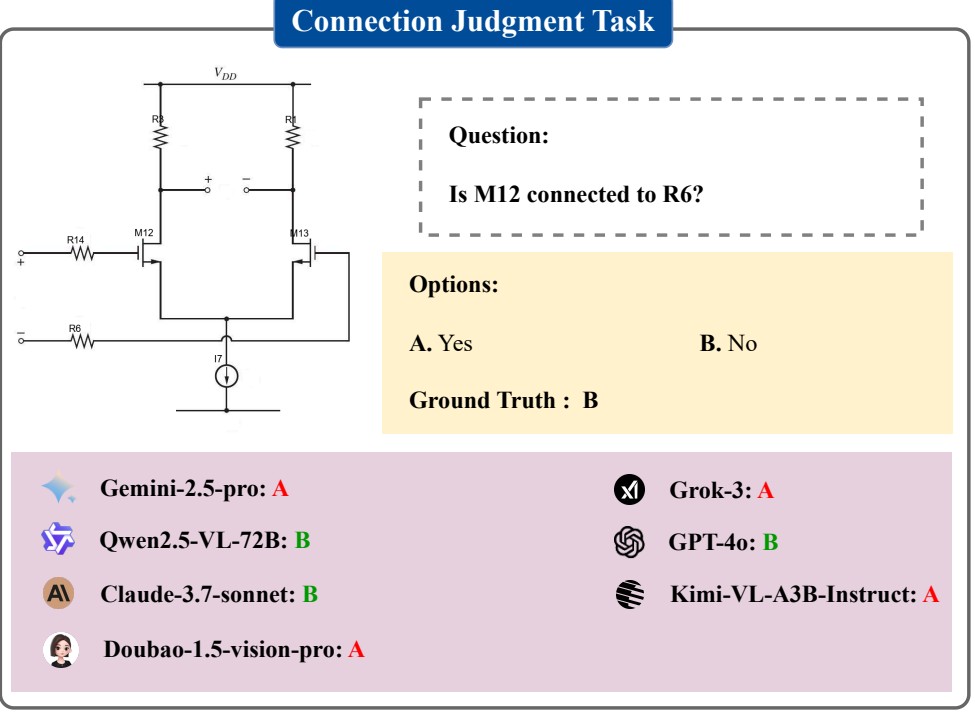

Figure 22: Example of **Connection Judgment** task across models

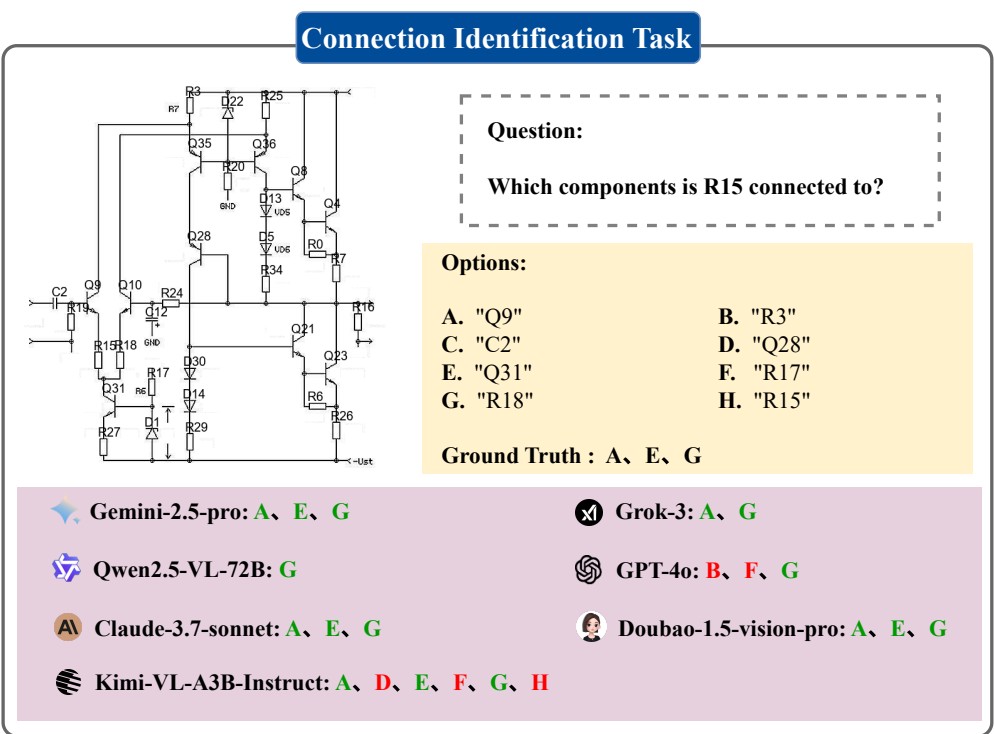

Figure 23: Example of **Connection Identification** task across models

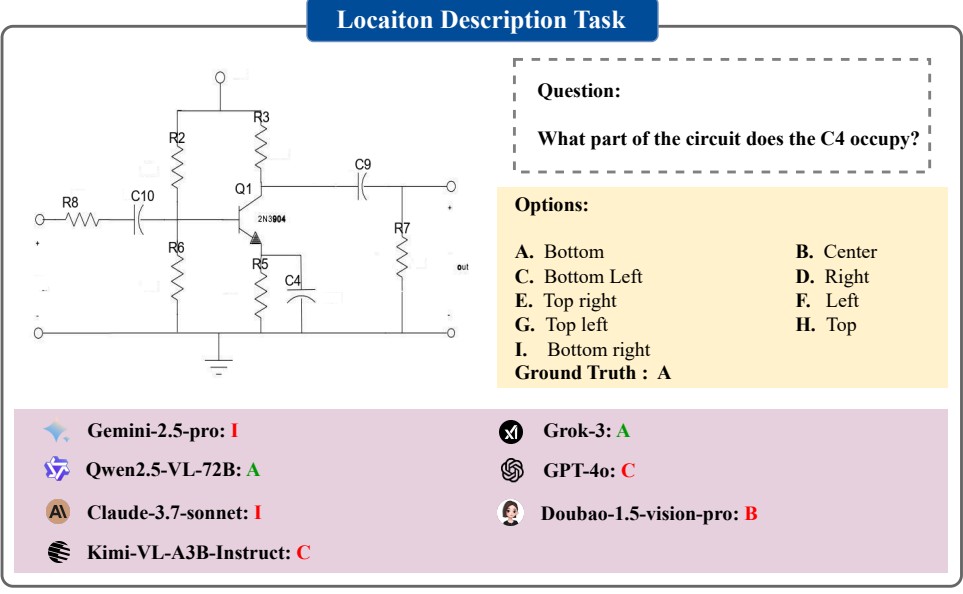

Figure 24: Example of **Location Description** task across models

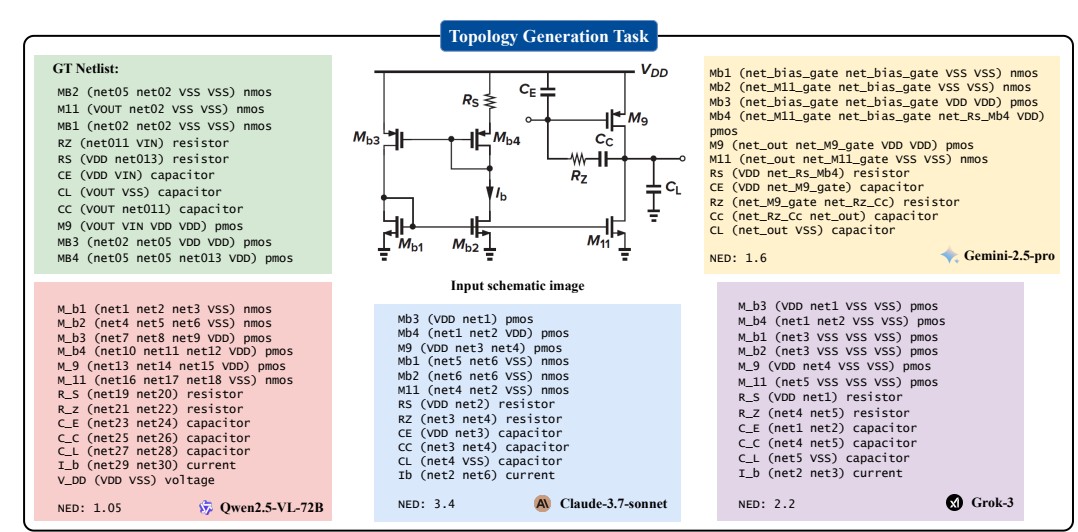

**Topology Generation Task**

**GT Netlist:**

```
MB2 (net05 net02 VSS VSS) nmos
M11 (VOUT net02 VSS VSS) nmos
MB1 (net02 net02 VSS VSS) nmos
RZ (net011 VIN) resistor
RS (VDD net013) resistor
CE (VDD VIN) capacitor
CL (VOUT VSS) capacitor
CC (VOUT net011) capacitor
M9 (VOUT VIN VDD VDD) pmos
MB3 (net02 net05 VDD VDD) pmos
MB4 (net05 net05 net013 VDD) pmos
```

**Input schematic image**

```
Mb1 (net_bias_gate net_bias_gate VSS VSS) nmos
Mb2 (net_M11_gate net_bias_gate VSS VSS) nmos
Mb3 (net_bias_gate net_bias_gate VDD VDD) pmos
Mb4 (net_M11_gate net_bias_gate net_Rs_Mb4 VDD)
pmos
M9 (net_out net_M9_gate VDD VDD) pmos
M11 (net_out net_M11_gate VSS VSS) nmos
Rs (VDD net_Rs_Mb4) resistor
CE (VDD net_M9_gate) capacitor
Rz (net_M9_gate net_Rz_Cc) resistor
Cc (net_Rz_Cc net_out) capacitor
CL (net_out VSS) capacitor
```

NED: 1.6          ◆ **Gemini-2.5-pro**

```
M_b1 (net1 net2 net3 VSS) nmos
M_b2 (net4 net5 net6 VSS) nmos
M_b3 (net7 net8 net9 VDD) pmos
M_b4 (net10 net11 net12 VDD) pmos
M_9 (net13 net14 net15 VDD) pmos
M_11 (net16 net17 net18 VSS) nmos
R_S (net19 net20) resistor
R_z (net21 net22) resistor
C_E (net23 net24) capacitor
C_C (net25 net26) capacitor
C_L (net27 net28) capacitor
I_b (net29 net30) current
V_DD (VDD VSS) voltage
```

NED: 1.05          洨 **Qwen2.5-VL-72B**

```
Mb3 (VDD net1) pmos
Mb4 (net1 net2 VDD) pmos
M9 (VDD net3 net4) pmos
Mb1 (net5 net6 VSS) nmos
Mb2 (net6 net6 VSS) nmos
M11 (net4 net2 VSS) nmos
RS (VDD net2) resistor
RZ (net3 net4) resistor
CE (VDD net3) capacitor
CC (net3 net4) capacitor
CL (net4 VSS) capacitor
Ib (net2 net6) current
```

NED: 3.4          Ⓐ **Claude-3.7-sonnet**

```
M_b3 (VDD net1 VSS VSS) pmos
M_b4 (net1 net2 VSS VSS) pmos
M_b1 (net3 VSS VSS VSS) pmos
M_b2 (net3 VSS VSS VSS) pmos
M_9 (VDD net4 VSS VSS) pmos
M_11 (net5 VSS VSS VSS) pmos
R_S (VDD net1) resistor
R_Z (net4 net5) resistor
C_E (net1 net2) capacitor
C_C (net4 net5) capacitor
C_L (net5 VSS) capacitor
I_b (net2 net3) current
```

NED: 2.2          ⓧ **Grok-3**

Figure 25: Example of **Topology Generation** task across models

## D.2 ANALYSIS TASK FOR ERROR ANALYSIS

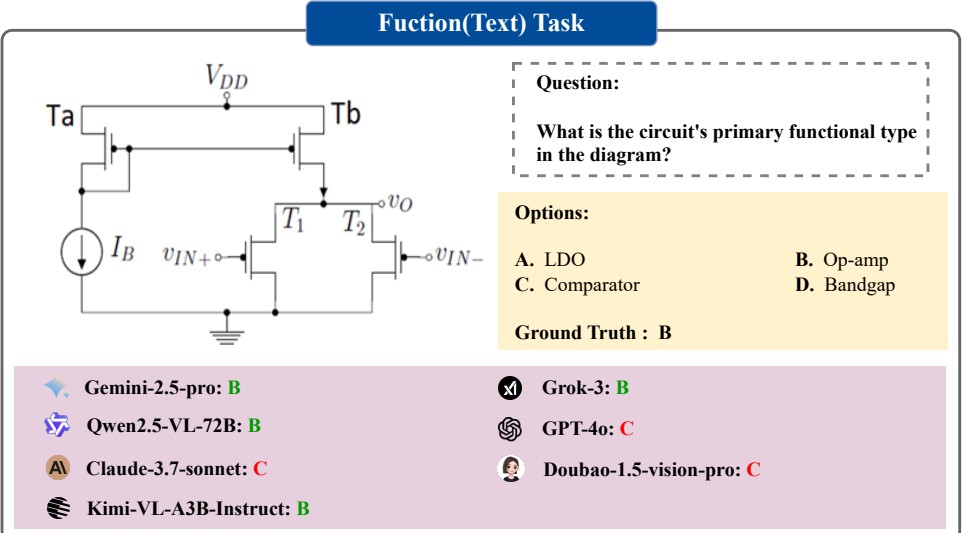

Figure 26: Example of **Function Text** task across models

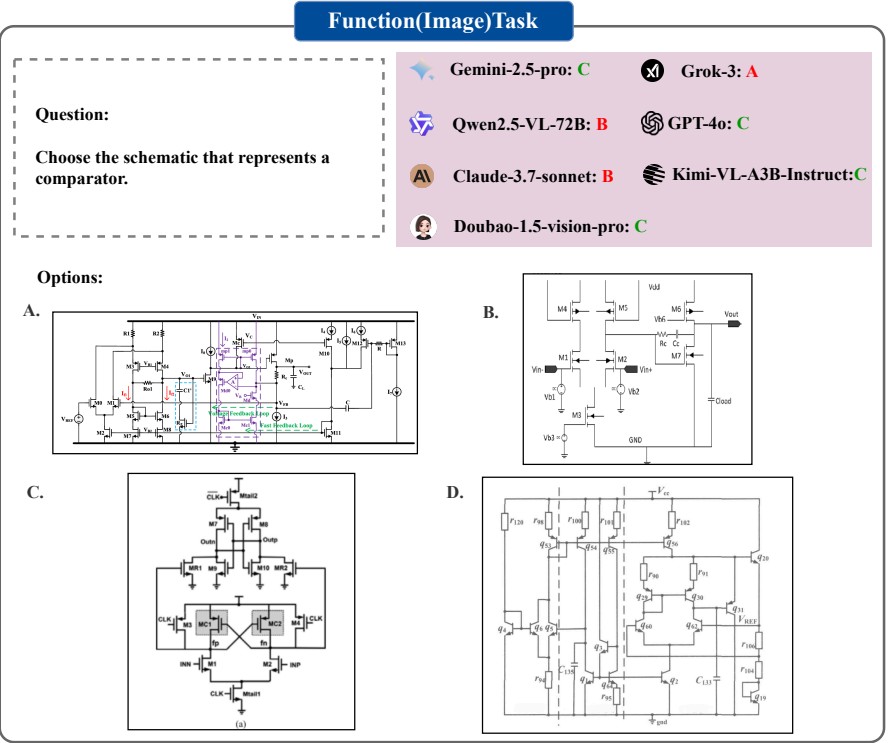

Figure 27: Example of **Function Image** task across models

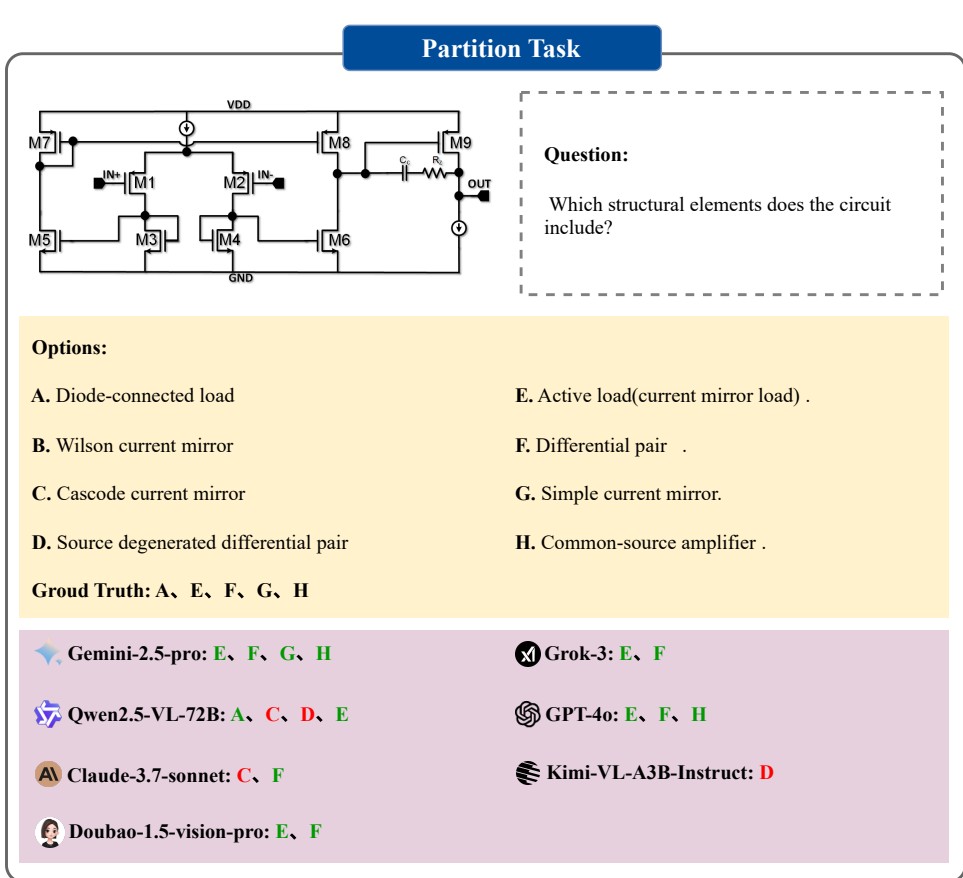

Figure 28: Example of **Partition Identification** task across models

**Reasoning Task**

VDD

Mp1  Mp2  $V_{b3}$

Mp3  Mp4  $V_{b4}$

Mn1  $V_{O1}$  $V_{O2}$

$V_{IN+}$  Mn2  $V_{IN-}$  $V_{b5}$

Mn3  Mn4

$V_{b1}$

Mn5  Mn6

$V_{b2}$

GND

**Question:**

What distinguishes this circuit as an operational amplifier in the given diagram?

**Ground Truth:**
This circuit qualifies as an operational amplifier because: The circuit includes a differential input stage formed by transistors Mn1 and Mn2, which is a fundamental feature of any operational amplifier. These transistors receive differential signals at their gates (VIN+ and VIN−), enabling differential signal amplification. Tail current sources are implemented using transistors Mp1 and Mp2, which help to set a proper operating point for Mp3-Mp4. The transistors Mn3-Mn6 form a cascode current source, indicated by the bias voltage Vb2 and Vb5, which can increase output impedance. A gain stage is built with transistors Mp3, Mp4, which function as a cascode and further amplify the signal. The outputs VO1 and VO2 suggest a differential output configuration, which is common in fully differential op-amp designs. Multiple bias voltages (Vb1, Vb2, Vb3, Vb4, and Vb5) are used to properly bias various stages of the circuit, which is a typical approach in analog integrated op-amp design to maintain correct operating points.

**Grok-3:**
The circuit is an operational amplifier primarily due to its differential pair Mn1 and Mn2, which amplify the input voltage difference through a resistive load formed by Mp3 and Mp4. A current source at the tail of the input stage, implemented by Mn5 and Mn6 under Vb2 control, sets the quiescent current. The cascode stage Mp1 and Mp2 provides voltage gain by stacking transistors, while Mp5 and Mp6 serve as active loads for the output stage, producing the differential outputs VO1 and VO2. The bias voltages (Vb1–Vb5) are critical for adjusting the transconductance of the input stage and tuning the cascode's gain, which aligns with conventional op-amp architectures that prioritize high open-loop gain and differential signal processing. (×)

**Claude-3.7-sonnet:**
This circuit qualifies as an operational amplifier because: The circuit includes a differential input stage formed by transistors Mn1 and Mn2, which is a fundamental feature of any operational amplifier. These transistors receive differential signals at their gates (VIN+ and VIN−), enabling differential signal amplification. Tail current sources are implemented using transistors Mp1 and Mp2, which help to set a proper operating point for Mp3-Mp4. The transistors Mn3-Mn6 form a cascode current source, indicated by the bias voltage Vb2 and Vb5, which can increase output impedance. A gain stage is built with transistors Mp3, Mp4, which function as a cascode and further amplify the signal. The outputs VO1 and VO2 suggest a differential output configuration, which is common in fully differential op-amp designs. Multiple bias voltages (Vb1, Vb2, Vb3, Vb4, and Vb5) are used to properly bias various stages of the circuit, which is a typical approach in analog integrated op-amp design to maintain correct operating points. (√)

**Gemini-2.5-pro:**
This circuit qualifies as an operational amplifier because: The circuit includes a differential input stage formed by transistors Mn1 and Mn2, which is a fundamental feature of any operational amplifier. These transistors receive differential signals at their gates (VIN+ and VIN−), enabling differential signal amplification. Tail current sources are implemented using transistors Mp1 and Mp2, which help to set a proper operating point for Mp3-Mp4. The transistors Mn3-Mn6 form a cascode current source, indicated by the bias voltage Vb2 and Vb5, which can increase output impedance. A gain stage is built with transistors Mp3, Mp4, which function as a cascode and further amplify the signal. The outputs VO1 and VO2 suggest a differential output configuration, which is common in fully differential op-amp designs. Multiple bias voltages (Vb1, Vb2, Vb3, Vb4, and Vb5) are used to properly bias various stages of the circuit, which is a typical approach in analog integrated op-amp design to maintain correct operating points. (√)

**Qwen2.5-VL-72B:**
This circuit qualifies as an operational amplifier because: The circuit includes a differential input stage formed by transistors Mn1 and Mn2, which is a fundamental feature of any operational amplifier. These transistors receive differential signals at their gates (VIN+ and VIN−), enabling differential signal amplification. Tail current sources are implemented using transistors Mp1 and Mp2, which help to set a proper operating point for Mp3-Mp4. The transistors Mn3-Mn6 form a cascode current source, indicated by the bias voltage Vb2 and Vb5, which can increase output impedance. A gain stage is built with transistors Mp3, Mp4, which function as a cascode and further amplify the signal. The outputs VO1 and VO2 suggest a differential output configuration, which is common in fully differential op-amp designs. Multiple bias voltages (Vb1, Vb2, Vb3, Vb4, and Vb5) are used to properly bias various stages of the circuit, which is a typical approach in analog integrated op-amp design to maintain correct operating points. (√)

**GPT-4o:**
The circuit is an operational amplifier primarily due to its differential pair Mn1 and Mn2, which amplify the input voltage difference through a resistive load formed by Mp3 and Mp4. A current source at the tail of the input stage, implemented by Mn5 and Mn6 under Vb2 control, sets the quiescent current. The cascode stage Mp1 and Mp2 provides voltage gain by stacking transistors, while Mp5 and Mp6 serve as active loads for the output stage, producing the differential outputs VO1 and VO2. The bias voltages (Vb1–Vb5) are critical for adjusting the transconductance of the input stage and tuning the cascode's gain, which aligns with conventional op-amp architectures that prioritize high open-loop gain and differential signal processing. (×)

**Kimi-VL-A3B-Instruct:**
The circuit is an operational amplifier primarily due to its differential pair Mn1 and Mn2, which amplify the input voltage difference through a resistive load formed by Mp3 and Mp4. A current source at the tail of the input stage, implemented by Mn5 and Mn6 under Vb2 control, sets the quiescent current. The cascode stage Mp1 and Mp2 provides voltage gain by stacking transistors, while Mp5 and Mp6 serve as active loads for the output stage, producing the differential outputs VO1 and VO2. The bias voltages (Vb1–Vb5) are critical for adjusting the transconductance of the input stage and tuning the cascode's gain, which aligns with conventional op-amp architectures that prioritize high open-loop gain and differential signal processing. (×)

**Doubao-1.5-vision-pro:**
This circuit qualifies as an operational amplifier because: The circuit includes a differential input stage formed by transistors Mn1 and Mn2, which is a fundamental feature of any operational amplifier. These transistors receive differential signals at their gates (VIN+ and VIN−), enabling differential signal amplification. Tail current sources are implemented using transistors Mp1 and Mp2, which help to set a proper operating point for Mp3-Mp4. The transistors Mn3-Mn6 form a cascode current source, indicated by the bias voltage Vb2 and Vb5, which can increase output impedance. A gain stage is built with transistors Mp3, Mp4, which function as a cascode and further amplify the signal. The outputs VO1 and VO2 suggest a differential output configuration, which is common in fully differential op-amp designs. Multiple bias voltages (Vb1, Vb2, Vb3, Vb4, and Vb5) are used to properly bias various stages of the circuit, which is a typical approach in analog integrated op-amp design to maintain correct operating points. (√)

Figure 29: Example of **Reasoning** task across models

**Caption Generation Task**

**Question:**

Give a detailed description of the following circuit.

**Ground Truth:**
An example of CMFB with MOS transistors in the linear region is shown in this diagram. It is a high-speed amplifier. A GBW of $850\mathrm{MHz}$ in two $5\mathrm{pF}$ capacitors can be reached thanks to the large currents, despite the old $1.2\upmu\mathrm{m}\mathrm{CMOS}$ technology. It is a folded cascode for a differential operation. The only additional feature is the feedforward around the slower PMOS cascode through capacitors $\mathrm{C_{f}}$. The outputs are around $_{0\mathrm{~V~}}$, because the gates of the NMOSs providing DC current to the input pair are at zero ground.

**Grok-3:**
This high-speed folded cascode amplifier employs a $1.2\upmu\mathrm{m}$ CMOS process to achieve a GBW of $850\mathrm{MHz}$ with $5\mathrm{pF}$ loads, primarily due to the low threshold voltages of the transistors. The CMFB circuit uses NMOS devices in the triode region to adjust the tail current, while feedforward capacitors $\mathrm{C_{f}}$ are connected around the NMOS cascode transistors to bypass their high-impedance nodes. The output DC level is stabilized at $0\mathrm{~V}$ because the PMOS current mirror gates are referenced to ground, ensuring proper biasing. The design avoids slow PMOS cascode by relying on dynamic current steering through the feedforward path. (×)

**Claude-3.7-sonnet:**
An example of CMFB with MOS transistors in the linear region is shown in this dirgram. It is a high-speed amplifier. A GBW of $850\mathrm{MHz}$ in two $5\mathrm{pF}$ capacitors can be reached thanks to the large currents, despite the a folded cascode for a differential operation. The only additional feature is the feedforward around the slower PMOS cascode through capacitors $\mathrm{C_{f}}$. The outputs are around $_{0\mathrm{~V~}}$, because the gates of the NMOSs providing DC current to the input pair are at zero ground. (√)

**Gemini-2.5-pro:**
An example of CMFB with MOS transistors in the linear region is shown in this dirgram. It is a high-speed amplifier. A GBW of $850\mathrm{MHz}$ in two $5\mathrm{pF}$ capacitors can be reached thanks to the large currents, despite the old $1.2\upmu\mathrm{m}\mathrm{CMOS}$ technology. It is a folded cascode for a differential operation. The only additional feature is the feedforward around the slower PMOS cascode through capacitors $\mathrm{C_{f}}$. The outputs are around $_{0\mathrm{~V~}}$, because the gates of the NMOSs providing DC current to the input pair are at zero ground. (√)

**Qwen2.5-VL-72B:**
This high-speed folded cascode amplifier employs a $1.2\upmu\mathrm{m}$ CMOS process to achieve a GBW of $850\mathrm{MHz}$ with $5\mathrm{pF}$ loads, primarily due to the low threshold voltages of the transistors. The CMFB circuit uses NMOS devices in the triode region to adjust the tail current, while feedforward capacitors $\mathrm{C_{f}}$ are connected around the NMOS cascode transistors to bypass their high-impedance nodes. The output DC level is stabilized at $0\mathrm{~V}$ because the PMOS current mirror gates are referenced to ground, ensuring proper biasing. The design avoids slow PMOS cascode by relying on dynamic current steering through the feedforward path. (×)

**GPT-4o:**
This high-speed folded cascode amplifier employs a $1.2\upmu\mathrm{m}$ CMOS process to achieve a GBW of $850\mathrm{MHz}$ with $5\mathrm{pF}$ loads, primarily due to the low threshold voltages of the transistors. The CMFB circuit uses NMOS devices in the triode region to adjust the tail current, while feedforward capacitors $\mathrm{C_{f}}$ are connected around the NMOS cascode transistors to bypass their high-impedance nodes. The output DC level is stabilized at $0\mathrm{~V}$ because the PMOS current mirror gates are referenced to ground, ensuring proper biasing. The design avoids slow PMOS cascode by relying on dynamic current steering through the feedforward path. (×)

**Kimi-VL-A3B-Instruct:**
This circuit implements a folded cascode operational amplifier optimized for high-speed performance, achieving a GBW of $850\mathrm{MHz}$ with two $5\mathrm{pF}$ capacitors due to the use of advanced biasing techniques in a $1.2\upmu\mathrm{m}$ CMOS process. The common-mode feedback (CMFB) mechanism employs NMOS transistors operating in the saturation region to stabilize the output common-mode voltage. A key feature is the feedforward path through capacitors $\mathrm{C_{f}}$ bypassing the slower NMOS cascode transistors, which enhances phase margin. The output DC level is maintained near $0\mathrm{~V}$ because the gates of the PMOS current sources are biased at the supply voltage, ensuring symmetric current distribution. (×)

**Doubao-1.5-vision-pro:**
This high-speed folded cascode amplifier employs a $1.2\upmu\mathrm{m}$ CMOS process to achieve a GBW of $850\mathrm{MHz}$ with $5\mathrm{pF}$ loads, primarily due to the low threshold voltages of the transistors. The CMFB circuitry uses NMOS devices in the triode region to adjust the tail current, while feedforward capacitors $\mathrm{C_{f}}$ are connected around the NMOS cascode transistors to bypass their high-impedance nodes. The output DC level is stabilized at $0\mathrm{~V}$ because the PMOS current mirror gates are referenced to ground, ensuring proper biasing. The design avoids slow PMOS cascodes by relying on dynamic current steering through the feedforward path. (×)

Figure 30: Example of **Caption Generation** task across models

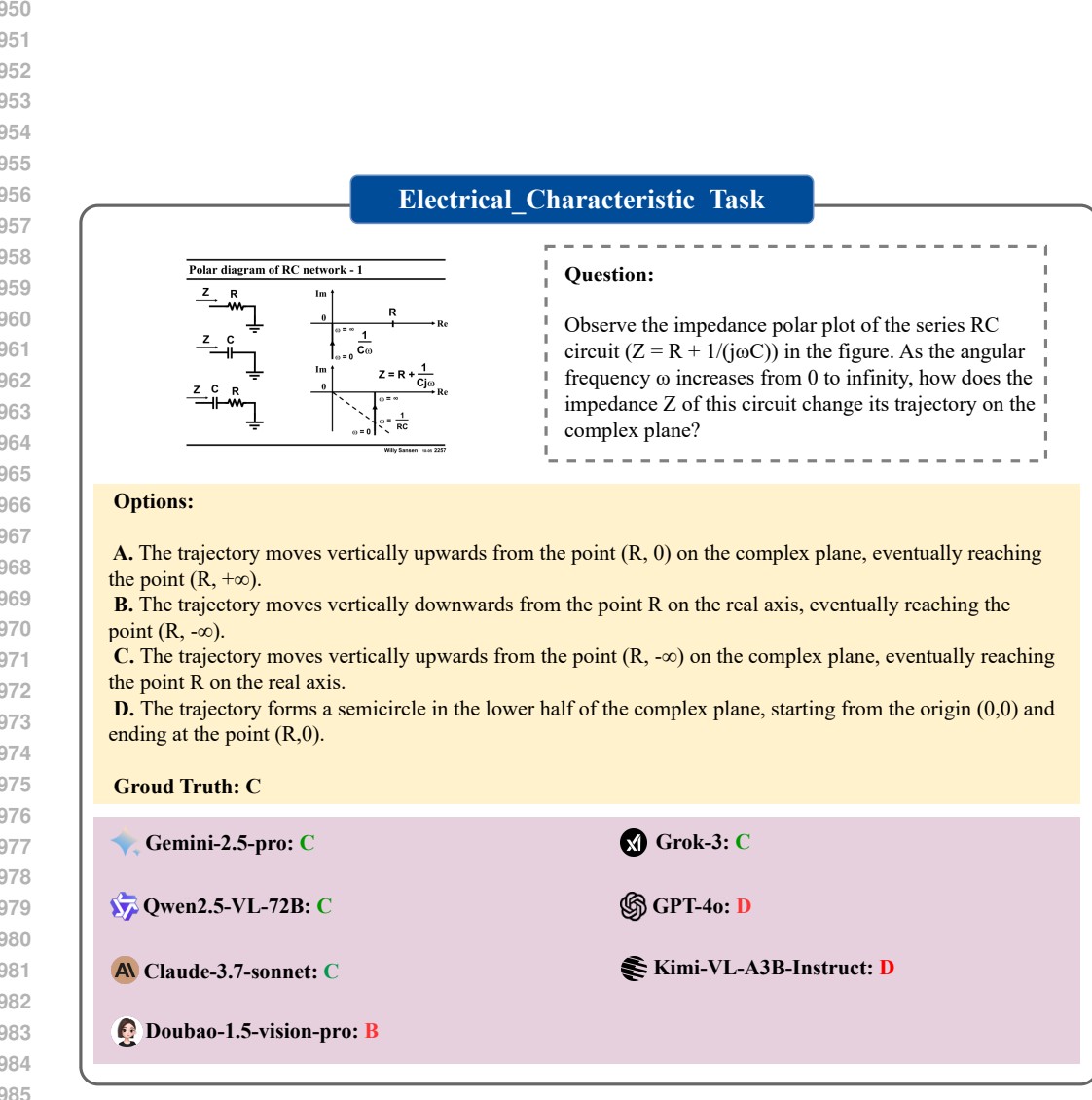

Figure 31: Example of **ElectricalCharacteristic** task across models

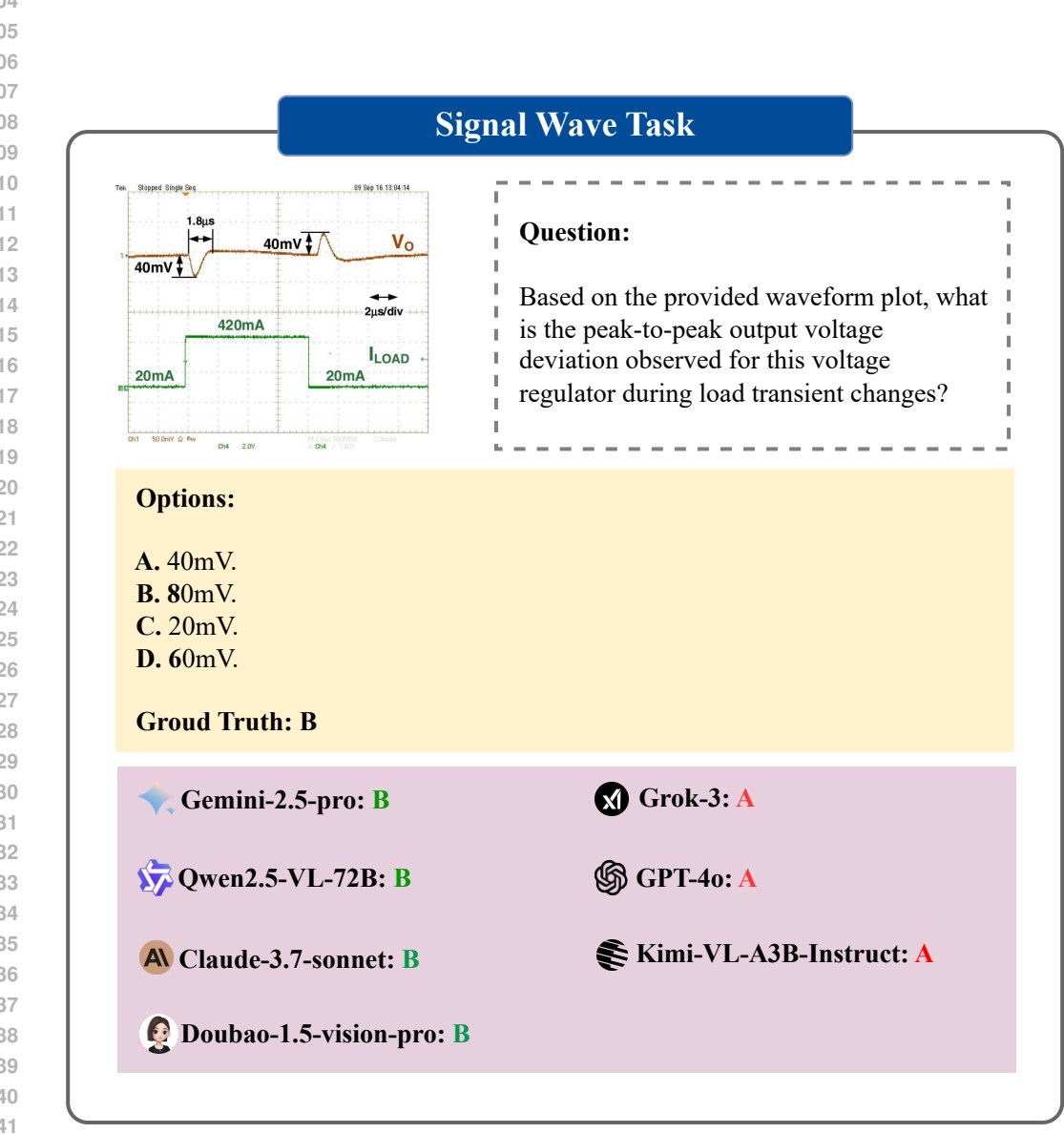

Figure 32: Example of **Signal Wave** task across models

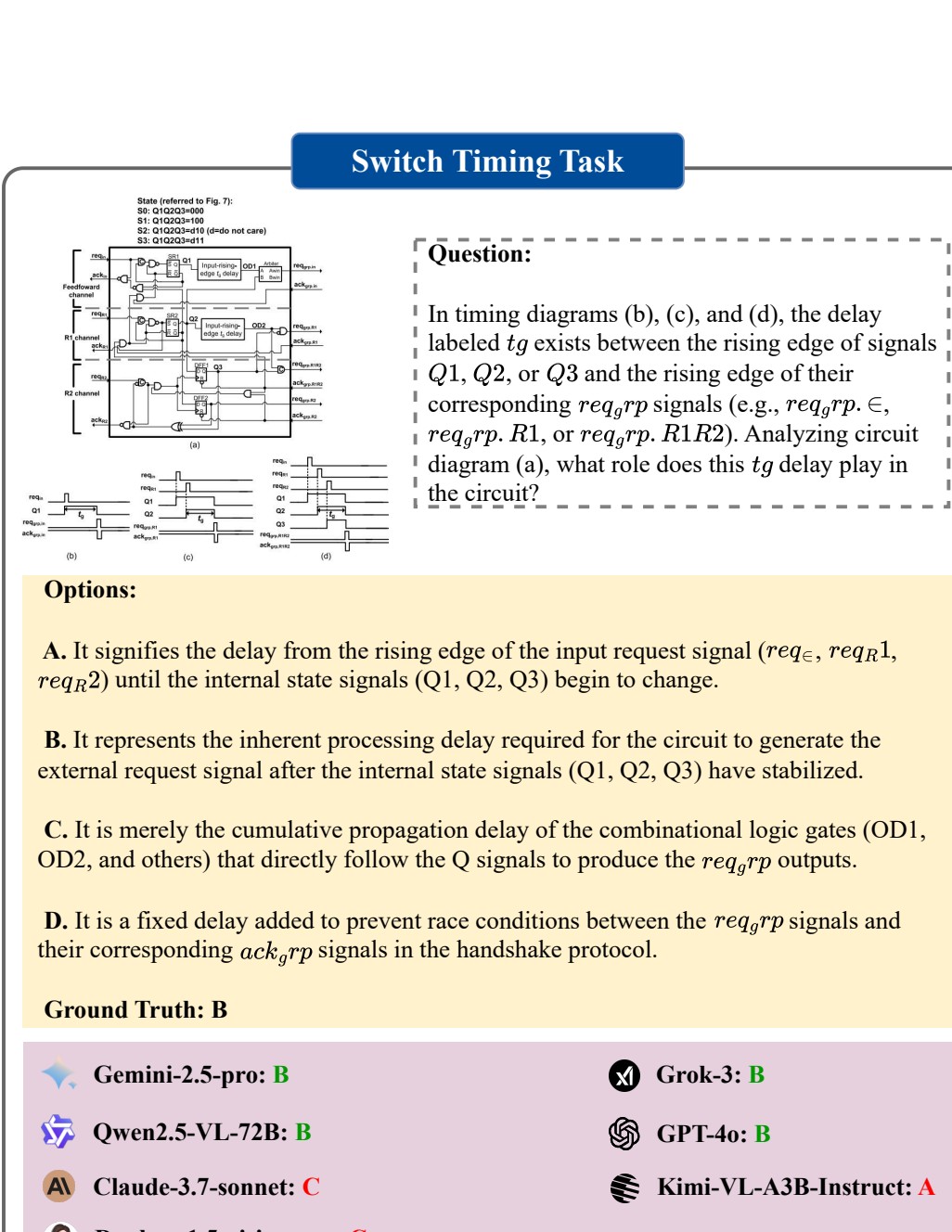

## Switch Timing Task

**Question:**

In timing diagrams (b), (c), and (d), the delay labeled $tg$ exists between the rising edge of signals $Q1$, $Q2$, or $Q3$ and the rising edge of their corresponding $req_grp$ signals (e.g., $req_grp.\in$, $req_grp.R1$, or $req_grp.R1R2$). Analyzing circuit diagram (a), what role does this $tg$ delay play in the circuit?

**Options:**

**A.** It signifies the delay from the rising edge of the input request signal ($req_\in$, $req_R1$, $req_R2$) until the internal state signals (Q1, Q2, Q3) begin to change.

**B.** It represents the inherent processing delay required for the circuit to generate the external request signal after the internal state signals (Q1, Q2, Q3) have stabilized.

**C.** It is merely the cumulative propagation delay of the combinational logic gates (OD1, OD2, and others) that directly follow the Q signals to produce the $req_grp$ outputs.

**D.** It is a fixed delay added to prevent race conditions between the $req_grp$ signals and their corresponding $ack_grp$ signals in the handshake protocol.

**Ground Truth: B**

**Gemini-2.5-pro: B**    **Grok-3: B**

**Qwen2.5-VL-72B: B**    **GPT-4o: B**

**Claude-3.7-sonnet: C**    **Kimi-VL-A3B-Instruct: A**

**Doubao-1.5-vision-pro: C**

Figure 33: Example of **Switching Timing** task across models

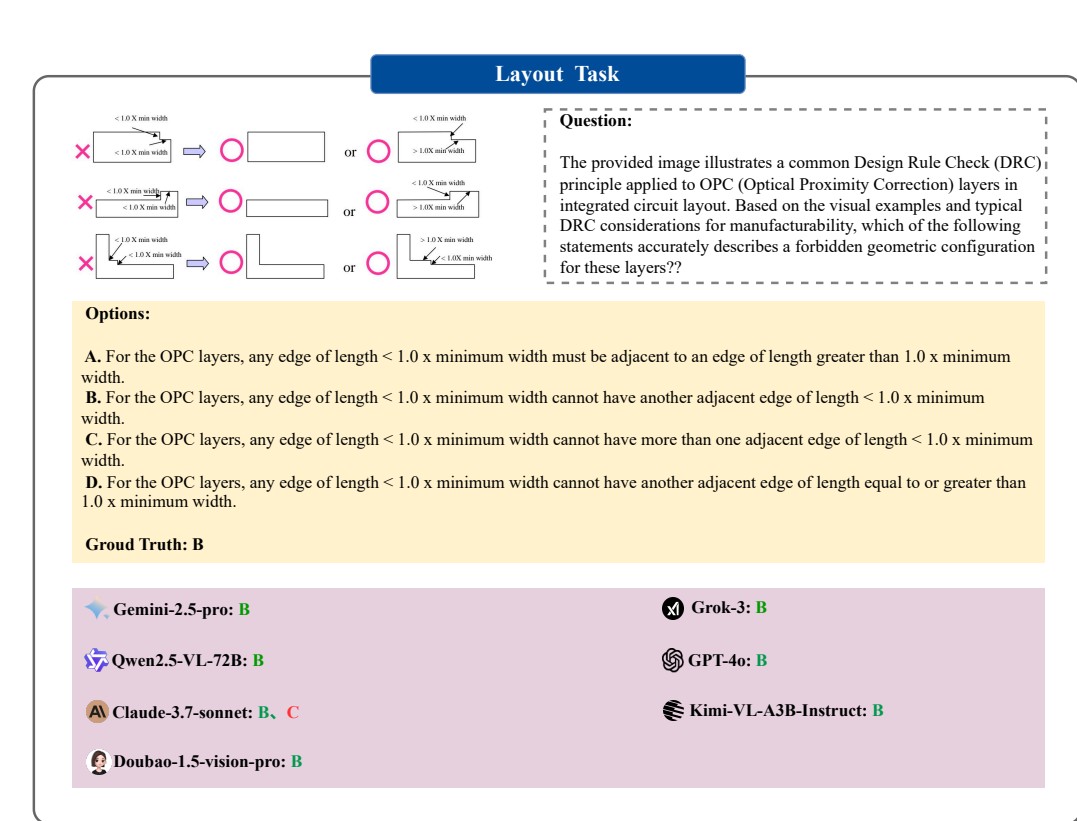

Figure 34: Example of **Layout** task across models

### D.3 DESIGN TASK FOR ERROR ANALYSIS

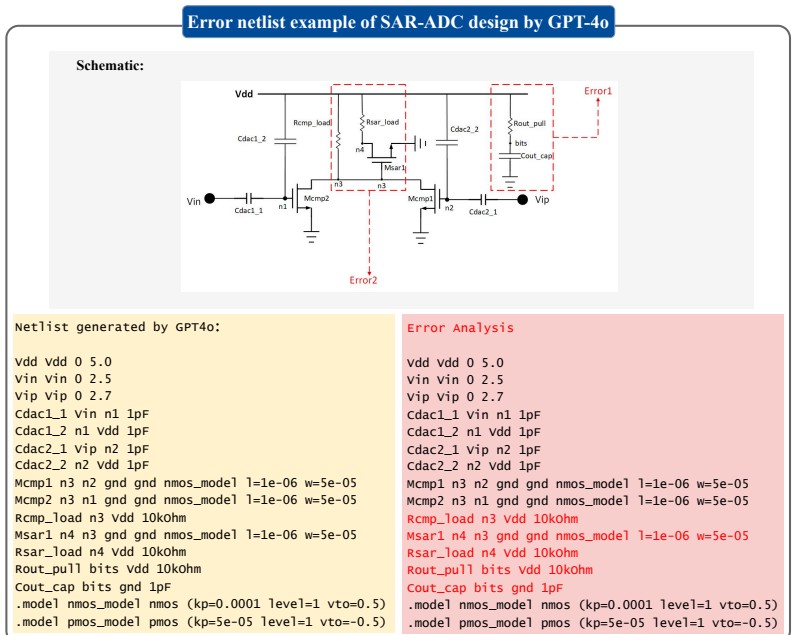

Figure 35: Example of **Circuit Design** task by GPT-4o

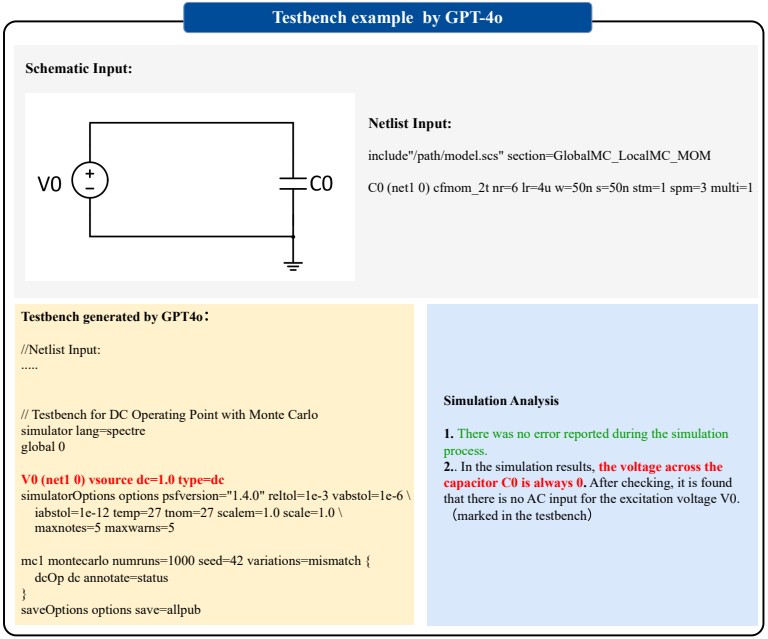

Figure 36: Example of **Testbench Design** task by GPT-4o

## D.4 EXAMPLE OF DEVICE GROUNDING

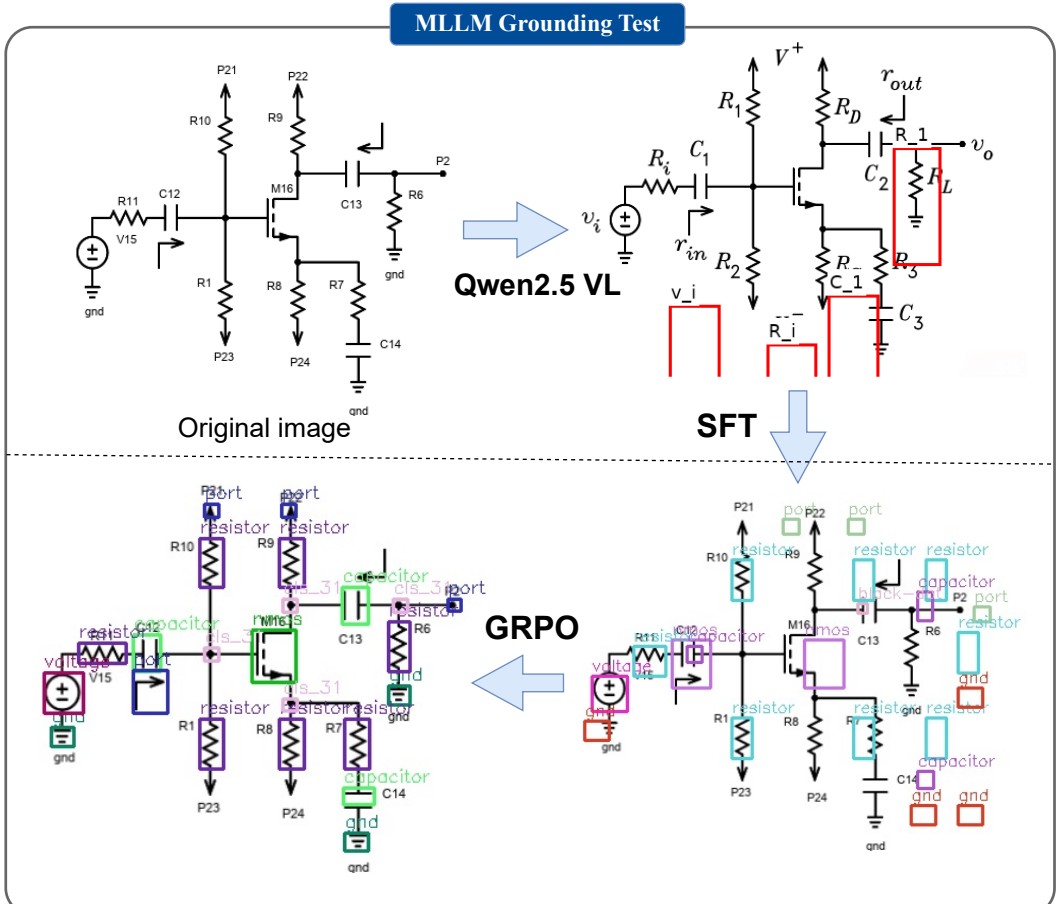

Figure 37: Example of Post-training with Qwen on device grounding.