# OpenReview forum: "AMSbench: A Comprehensive Benchmark for Evaluating MLLM Capabilities in AMS Circuits"
_ICLR.cc/2026/Conference — ICLR 2026 Conference Withdrawn Submission_

### Official Review · Reviewer_YmmK · 2025-10-20

**Soundness:** 3
**Presentation:** 3
**Contribution:** 2
**Rating:** 4
**Confidence:** 4

**Summary:**

This paper proposes AMSbench, a benchmark comprising approximately 8,000 questions designed to evaluate MLLMs on tasks such as schematic perception and circuit analysis. The authors assess models including Qwen 2.5-VL and Gemini 2.5 Pro. The results reveal major limitations in current MLLMs’ multi-modal reasoning and circuit design capabilities, underscoring the need for stronger circuit-specific understanding to advance automated AMS design.

**Strengths:**

1. The benchmark is comprehensive and well-structured.

2. The paper is clearly written and easy to follow.

3. Analog/Mixed-Signal (AMS) circuits understanding for MLLMs is an important problem.

**Weaknesses:**

1. Missing related work: The paper does not discuss EEE-Bench [1], a comprehensive multimodal electrical and electronics engineering benchmark that also focuses heavily on circuit analysis.

2. Overstated claims and unclear novelty: The authors claim that AMSbench is the first holistic benchmark for systematically evaluating MLLM performance in AMS circuits. However, prior works such as EEE-Bench and MMCircuitEval have already conducted extensive studies in this domain. The paper should clearly articulate how AMSbench differs from these benchmarks and justify its unique contributions. Furthermore, it should explain why existing benchmarks are insufficient for the intended evaluation goals.

3. Vague data collection and curation details: The Data Collection and Curation section lacks sufficient detail. The authors briefly mention using “carefully crafted prompt engineering and filter strategies” but provide no specifics about these methods. Similar issues appear in the question–answer (QA) creation process, which is also insufficiently described.

4. Unclear evaluation methodology: The evaluation process is not well explained. It remains unclear how model responses were parsed to extract answers or how accuracy was computed. If these details are included elsewhere, the authors should provide clearer references.

5. Limited evaluation scope: The study only evaluates seven models. Including additional models such as o3, GPT-4.1, Qwen2.5-VL series, and InternVL-3 series would provide a more comprehensive assessment.

6. Unconvincing discussion in Section 5: The discussion section (Section 5) lacks both qualitative and quantitative evidence. Without empirical validation, the arguments presented remain speculative and unpersuasive.

[1] EEE-Bench: A Comprehensive Multimodal Electrical and Electronics Engineering Benchmark. CVPR 2025.

**Questions:**

See weaknesses.

I am happy to revise my score if authors can address my concerns.

---

### Official Review · Reviewer_N3HC · 2025-10-30

**Soundness:** 3
**Presentation:** 2
**Contribution:** 3
**Rating:** 4
**Confidence:** 4

**Summary:**

This work proposes a benchmark, called AMSbench, for the goal of assessing model capabilities across diverse analog/mixed-signal (AMS) circuits.
It spans different tasks including circuit schematic perception, circuit analysis and circuit design.
It comprises approximated 8k question-answer pairs with difficult levels.
Besides, both open-source and proprietary models are considered during the evaluation.

**Strengths:**

- This work focuses on a meaning direction: building a benchmark for AMS circuits, which may have great impact for the future research.
- The design of benchmark is based on a deep and comprehensive understanding of circuit design.
- There are extensive experiments, which offers lots of insight to the audience.

**Weaknesses:**

- What is the error rate in the proposed benchmark? As mentioned, the questions and answers are created by humans and AI models, will there be any error in the benchmark (e.g., the answer is incorrect or the question is not a reasonable one)? Do we have a mechanism in the data curation to control the error rate?
- In lines 213 to 214, it is mentioned that there are textual question answering (TQA) in the benchmark. In Section 4.2, the metrics considered include ACC, F1, NED and pass@k. It seems that all of them cannot be used to evaluate TQA. How is TQA evaluated?
-  For design task, how are difficult levels decided? There is only a very vague explanation for it (i.e., complexity in line 284).
- Besides the number of QA pairs, how many original circuits (which may be in any formats) are involved? I guess for a certain circuit, it may have more than more one questions, right?

**Questions:**

see weaknesses.

---

### Official Review · Reviewer_jvcy · 2025-10-31

**Soundness:** 3
**Presentation:** 3
**Contribution:** 2
**Rating:** 4
**Confidence:** 3

**Summary:**

This paper introduces AMSbench, a large-scale multimodal benchmark for evaluating multimodal large language models (MLLMs) in Analog/Mixed-Signal (AMS) circuit understanding and design.
AMSbench systematically covers three key dimensions of AMS workflows: schematic perception, circuit analysis, and circuit design, comprising about 8,000 question-answer pairs. The benchmark includes schematic-to-netlist recognition, waveform interpretation, functional reasoning, and testbench generation tasks.
The authors benchmark several MLLMs (GPT-4o, Gemini-2.5 Pro, Claude-Sonnet, Grok-3, Qwen2.5-VL, etc.), and reveal that while models handle basic component recognition and schematic captions well, they perform poorly on netlist reconstruction and testbench synthesis. The study concludes that current MLLMs lack fine-grained perception and domain-specific reasoning needed for fully automated AMS circuit design.

**Strengths:**

- Comprehensive Benchmark Scope and Design
AMSbench is the first end-to-end evaluation pipeline that includes schematic perception, circuit analysis, and design reasoning all in one benchmark.  Prior research (e.g., MMCircuitEval, AnalogCoder) were primarily concerned with symbolic circuit reasoning or schematic captioning, but AMSbench expressly combines multimodal visual and textual information.
Because of its broad reach, it is ideally positioned to assess both low-level perception (component location, connection tracing) and high-level thinking (design intent, testbench logic generation) at the same time.

- Fine-grained metric and diagnostic analysis
The authors go beyond accuracy measurements by reporting Netlist Edit Distance for structural correctness, Functional Equivalence Ratio for schematic reasoning, and pass@k for design generation activities.

- Evaluation of Different MLLMs
The inclusion of several prominent MLLMs, including closed- and open-source models (GPT-4o, Gemini 2.5, Claude Sonnet, Grok-3, Qwen 2.5-VL, and others), provides a complete perspective of the field.  The work compares models fairly using comparable prompting settings, highlighting capacity disparities between commercial and research systems.

**Weaknesses:**

- Limited novelty in methodology
The benchmark-building workflow is mostly based on current dataset production paradigms such as template-driven QA, schematic parsing, and human verification.  The key contribution is domain adaption to AMS, not a methodological leap in benchmark construction.

- Imbalanced task distribution.
The perception subset accounts for around 6,000 samples, but the design and testbench generation subsets are much less (~ hundreds).  This mismatch reduces statistical confidence in conclusions about higher-level reasoning capabilities. A stratified sampling or data-balancing strategy could help ensure each subdomain contributes meaningfully to evaluation.

- Limited Evaluation Diversity and Depth.
Although eight MLLMs are evaluated, the most are general-purpose vision-language models.  The benchmark would be even more useful if it includes comparisons against domain-adapted or fine-tuned baselines (such as LLMs coupled with retrieval-augmented EDA agents).  Without them, it's hard to determine how much of the performance disparity is attributable to a lack of domain grounding vs core model architectural constraints.

- Incomplete Statistical Reporting
The analysis does not assess annotation consistency or inter-rater reliability, which could be critical for a benchmark based on human validation of domain-specific responses.

- Lack of Difficulty Calibration / Curriculum Analysis
Tasks range from simple schematic labeling to complicated circuit creation, however the complexity is not clearly stated.  An investigation comparing task difficulty with model performance (e.g., by component count or circuit hierarchy depth) would give more insight into scaling behavior.

**Questions:**

- Could the authors comment on more details about their annotation workflow? How many domain experts participated? How were differences resolved? Was inter-annotator agreement measured?

- Was the validity of reasoning-based queries  confirmed using any expert judgment? Clarifying this would increase trust in the dataset's dependability.

- The benchmark heavily prioritizes perception tasks (~6k samples) over design and synthesis tasks (~hundreds). How do the authors ensure representative coverage of AMS reasoning difficulty? Are there plans to balance future releases by expanding the underrepresented subsets? Also, are circuit categories (e.g., amplifiers, oscillators, converters) uniformly represented, or are certain families over-represented due to data availability?

- The paper references prior works such as MMCircuitEval and AnalogCoder but does not provide more detailed quantitative comparison. Did any subsets overlap in schematic or question type?

- Since AMS tasks are highly knowledge-intensive, have the authors considered evaluating argumented system such as retrieval-augmented MLLMs?

Overallm I think AMSbench is a relevant and well-executed benchmark paperthat demonstrates the limits of modern MLLMs in analog circuit reasoning.  While the approach is basic, the contribution's scope and depth make it a substantial contribution to the community. I'd like to consider raising the score if some questions can be improved through the paper.

---

### Official Review · Reviewer_B1TW · 2025-11-01

**Soundness:** 2
**Presentation:** 3
**Contribution:** 2
**Rating:** 4
**Confidence:** 4

**Summary:**

This paper proposes a new benchmark targeted at AMS circuit. It breaks down the specific task type by knowledge hierarchy, from understanding to designing; problems for each task are separated by their difficulty. Evaluation on advanced frontier models show that MLLMs lack the capability to understand AMS circuit and struggles more at design.

**Strengths:**

1. Coverage is comprehensive. This work covers AMS in depth, providing 8674 question-answer pairs spanning three core tasks—schematic perception, circuit analysis, and circuit design, which are further broken down into 18 distinct sub-tasks.
2. The generative evaluation is interesting to see, and it highlights a key drawback of MLLM at AMS. (although the relatie #of question for design is small, 68)
3. The paper is easy to follow.

**Weaknesses:**

1. The topic has been addressed before. This work seems very similar to [1], which also benchmarks MLLM for electrical circuits or related topics and have been published at prior conference. They also reached similar results (MLLM perform poorly, the domain gap in pretraining could be one cause, etc), however, it is not mentioned in this manuscript. The authors are encouraged to address this work and clearly clarify how they are different from this work.
2. Benchmark difficulty is in question. Besides generative task, the two other areas evaluated (perception and analysis) all seem pretty high performance from the top models already (in the high 70% to 90%+, even for some open models). This calls into question the longevity of this benchmark itself and undermines the potential usefulness in measuring and tracking the advancement of MLLM in this domain.



[1] Li. et al., EEE-Bench: A Comprehensive Multimodal Electrical And Electronics Engineering Benchmark

**Questions:**

1. The complete failure of testbench generation is surprising. The authors attribute this to a lack of pre-training data. I wonder if it could have been an artifact of the prompt or task format? Did you experiment with any few-shot learning or cot prompting (or other advanced prompting methods) for this task (and for other tasks as well), or were all evaluations zero-shot?

---

### Note · Authors · 2025-11-26

I have read and agree with the venue's withdrawal policy on behalf of myself and my co-authors.